# Systematic identification of non-coding pharmacogenomic landscape in cancer

Yue Wang[1], Zehua Wang[1], Jieni Xu[1], Jiang Li[1], Song Li[1], Min Zhang[1] & Da Yang[1,2,3]

Emerging evidence has shown long non-coding RNAs (lncRNAs) play important roles in cancer drug response. Here we report a lncRNA pharmacogenomic landscape by integrating multi-dimensional genomic data of 1005 cancer cell lines and drug response data of 265 anti-cancer compounds. Using Elastic Net (EN) regression, our analysis identifies 27,341 lncRNA-drug predictive pairs. We validate the robustness of the lncRNA EN-models using two independent cancer pharmacogenomic datasets. By applying lncRNA EN-models of 49 FDA approved drugs to the 5605 tumor samples from 21 cancer types, we show that cancer cell line based lncRNA EN-models can predict therapeutic outcome in cancer patients. Further lncRNA-pathway co-expression analysis suggests lncRNAs may regulate drug response through drug-metabolism or drug-target pathways. Finally, we experimentally validate that *EPIC1*, the top predictive lncRNA for the Bromodomain and Extra-Terminal motif (BET) inhibitors, strongly promotes iBET762 and JQ-1 resistance through activating MYC transcriptional activity.

[1] Center of Pharmacogenetics, Department of Pharmaceutical Sciences, University of Pittsburgh, Pittsburgh, PA 15261, USA. [2] University of Pittsburgh Cancer Institute, University of Pittsburgh, Pittsburgh, PA 15261, USA. [3] Department of Computational and System Biology, University of Pittsburgh, Pittsburgh, PA 15261, USA. Correspondence and requests for materials should be addressed to M.Z. (email: miz45@pitt.edu) or to D.Y. (email: dyang@pitt.edu)

Heterogeneous response of individuals to cancer therapies has been largely attributed to genetic difference of tumor cells[1]. Using cell-line-based panels, annotated with both genetic and pharmacological data, to gain insights into the mechanism of anti-cancer drug response has been considered as the cornerstone of precision cancer medicine[2]. Those large-scale high-throughput cancer pharmacogenomics efforts, mainly focusing on protein coding components of the genome, have led to many insightful discoveries[3–6] but also raised new questions: few new biomarkers and drivers were identified to fully explain the regulation of drug resistance in cancer[7].

Emerging evidence from large-scale studies, such as the Encyclopedia of DNA Elements (ENCODE), suggest that up to 80% of the human genome is capable of being transcribed into primary RNA transcripts, including numerous long non-coding RNA (lncRNA)[8,9]. These studies have identified 16,033 lncRNAs genes, which are ncRNAs larger than 200 nt and do not have protein-coding potential[10]. Further genome-wide characterization of the human cancer transcriptome revealed that lncRNAs are among the most prevalent transcriptional changes in cancer[11–13]. Similar to the protein-coding genes, lncRNAs can play critical roles in tumor initiation and progression[9,14–18], as well as cancer therapy response[19–21]. Large-scale cancer genome and pharma-cogenomics projects, such as The Cancer Genome Atlas (TCGA)[4], Cancer Cell Lines Encyclopedia (CCLE)[2], Genomics of Drug Sensitivity in Cancer (GDSC)[22], and Cancer Therapeutics Response Portal (CTRP)[23] have provided an unprecedented opportunity to systematically determine the regulatory roles of lncRNA in cancer drug response by generating RNA-seq data in conjunction with clinical and drug response data from thousands of tumor samples and cancer cell lines.

Here we integrated multiple dimensional pharmacogenomics data of 2614 cancer-related lncRNAs in 5605 primary tumor samples and 505 cancer cell lines from 27 cancer types to build lncRNA-based drug response models for 265 anti-cancer agents. We have demonstrated that cancer cell lines could recapitulate the lncRNAs alterations, i.e. expression, copy number and methylation aberrations, in primary tumors. Further Elastic Net (EN) regression analysis identified 27,341 lncRNA-drug predictive pairs in cancer cell lines. Notably, lncRNA-based EN models can predict chemotherapy response not only in independent cell line pharmacogenomic databases, but also in cancer patients. Mechanistically, our computational analysis and experimental validation reveal that lncRNAs may regulate cancer drug response through drug metabolism and drug-target pathways. To our best knowledge, this is the first study to systematically link noncoding genotypes with drug response phenotypes in both cancer cell lines and patient tumors.

## Results

### Recap of lncRNA alterations in primary tumors by cell lines.
To assess whether cancer cell lines resemble the primary tumors in the perspective of lncRNA alterations, we obtained RNA-seq, copy number and DNA methylation data in 5605 TCGA tumor samples and 505 cancer cell lines across 27 cancer types from GDSC and CCLE database. The 2614 cancer-related lncRNAs were first identified based on differential expression between patient tumors and normal tissues in the TCGA database (Methods section). Among the 2614 cancer-related lncRNAs derived from patient tumor samples, all of them are expressed in at least one cancer cell line; and 2511 (96.06%) are expressed in at least three cell lines (Fig. 1a, b). We further repurposed the Affymetrix SNP 6.0 microarray and Illumina 450K Human Methylation microarray to obtain the copy number and epigenetic alterations of cancer-related lncRNAs in each tumor sample

and cell line as previously described[24,25] (Fig. 1a, Methods section). The lncRNA alterations were significantly correlated between cell lines and patient tumors for 14 out of 18 (77.78%) cancer types based on expression, 15 out of 19 (78.94%) cancer types based on DNA methylation, and 13 out of 18 (72.22%) cancer types based on copy number alterations (Fig. 1c–e).

We next used a previously described nearest-neighbor matching algorithm[22] to determine whether lncRNA alteration profiles in cancer cell lines are representative of patient tumors based on the lncRNA alterations (Methods section). Within the top 5 nearest neighbors, the algorithm could 100% match the tissue of origin of cell lines to primary tumors using lncRNA expression with a random expectation of matching rate at 33.3%. This percentage is around 89.5% when using methylation and is 88.9% when using copy number with random expectation at 15.8% and 27.8%, respectively. After integrating three features, the success rate of matching is around 94.4% (random expectation at 22.2%) within the top 5 nearest neighbors (Fig. 1f, Supplementary Data 1). The concordance of lncRNA alterations between primary tumors and cancer cell lines was most prominent in the expression level, followed by DNA methylation and copy number alterations.

### A landscape of LncRNA-drug interaction in cancer cell lines.
LncRNAs expression profile and drug response data across 505 cancer cell lines were integrated to identify predictive lncRNA-drug pairs (Supplementary Fig. 1a, Methods section). For each cell line, the drug response data include the values of IC50 and area under the curve (AUC)[22] of 265 anti-cancer agents from the GDSC database (Fig. 2a, Supplementary Fig. 2a, Supplementary Data 2). By conjugating Elastic Net (EN) Regression and boot-strap aggregating, we built lncRNA-drug response prediction models for each agent across all the cell lines (pan-cancer model) or cell lines from a specific cancer type (cancer-specific model) (Methods section). The model performance was assessed by the correlation between the predicted response and the observed response for each agent. Overall, pan-cancer models for 265 drugs achieved median performance at $r = 0.31$ ($p = 6.76 \times 10^{-5}$, Pearson's correlation) in bootstrapping. Cancer-specific models built by smaller numbers of samples, on the other hand, achieved a decreased median performance at $r = 0.13$ (Pearson's correlation) (Fig. 2b).

To determine each lncRNA's contribution to drug response, a predictive score (PS) was assigned to each lncRNA based on the frequency it was selected by EN regression throughout the bootstrapping (Supplementary Fig. 1 b, Methods section). The lncRNA with higher PS would have better association with the corresponding agent response, which was defined as a lncRNA-drug predictive pair. Using IC50 as an indicator of drug response, this feature selection process identified 27,341 lncRNA-drug predictive pairs through pan-cancer modeling (on median 100 predictive lncRNAs per agent) (Fig. 2c, d, Supplementary Data 3, Methods section). When using AUC[22] as an indicator of drug response, a highly consistent lncRNA-drug pairs network was obtained ($r = 0.63$, $p < 0.0001$, Pearson's correlation) (Fig. 2e, Methods section).

To validate the identified lncRNA-drug predictive pairs, we calculated the correlation between the expression of these lncRNAs and drug response in two independent datasets: the CCLE and CTRP (Supplementary Data 2). We observed a significantly higher correlation for the lncRNA-drug predictive pairs compared to the non-predictive ones in both datasets (two-side Kolmogorov-Smirnov (KS) test: $p = 1.01 \times 10^{-7}$ for CCLE; $p = 2.39 \times 10^{-32}$ for CTRP) (Supplementary Fig. 2a). Moreover, we also performed the same feature selection procedure using the

drug response data from CCLE and CTRP datasets. For the 14 overlapping agents between CCLE and GDSC, we identified 512 and 1366 lncRNA-drug pairs respectively. Among the 512 lncRNA-drug pairs in CCLE, 90 (17.6%) were also found to be significant (PS >= 0.25) in GDSC (odds ratio (OR) = 5.71, $p = 1.41 \times 10^{-34}$ Fisher's exact test). For the 76 overlapping drugs

between CTRP and GDSC, we identified 4827 and 7938 lncRNA-drug pairs, respectively. Among the 4827 lncRNA-drug pairs in CTRP, 612 (12.7%) were also found to be significant (PS >= 0.25) in GDSC (OR = 3.59, $p = 8.89 \times 10^{-136}$, Fisher's exact test). Notably, the lncRNA-drug pairs identified by EN regression model have a significantly higher robustness among independent

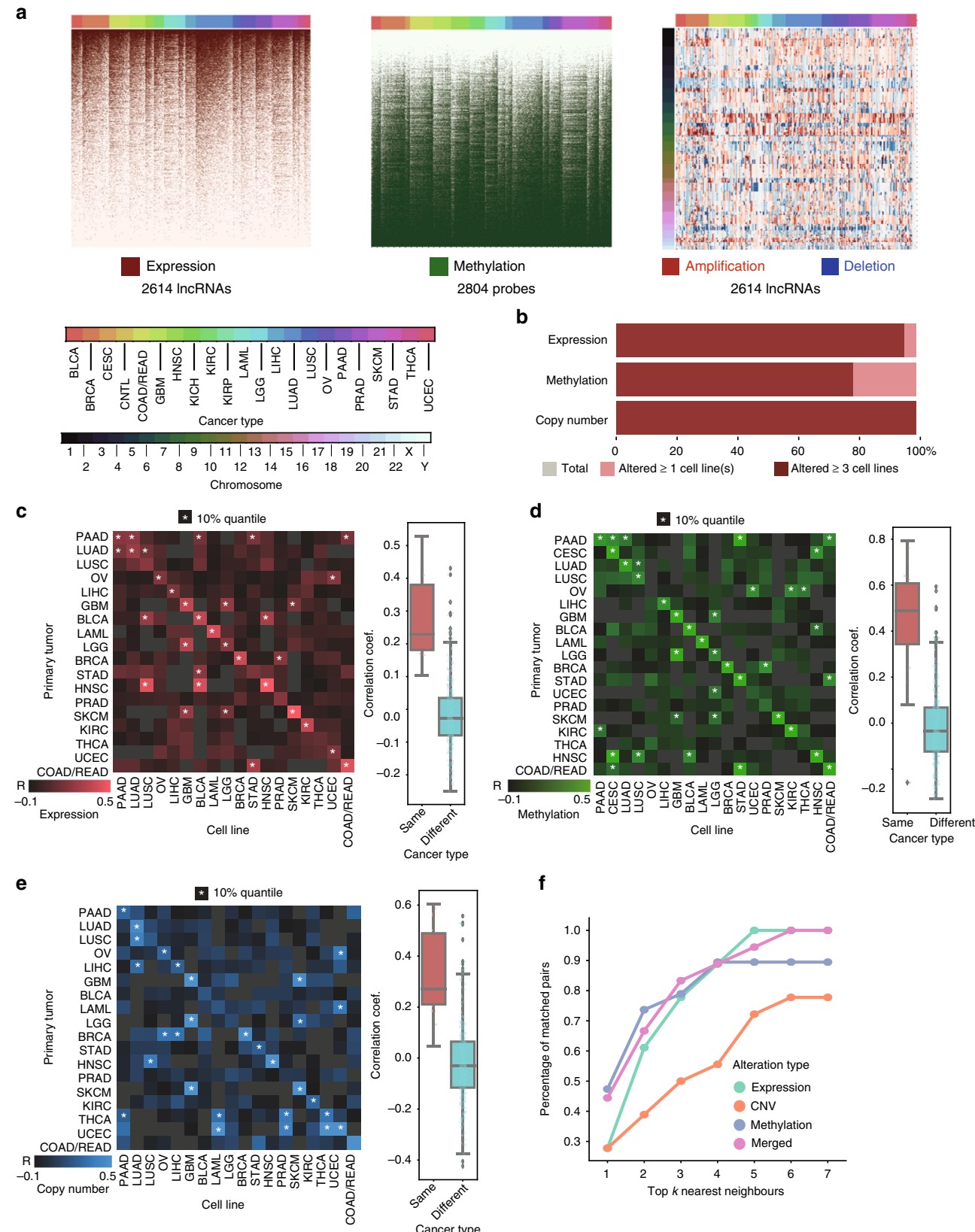

databases than those identified by Spearman's correlation (3.6% for CCLE and 1.4% for CTRP).

The EN regression successfully identified lncRNAs that are well documented to regulate drug response. For instance, our model identified *MEG3* overexpression as a predictor of cisplatin sensitivity, which is consistent with previous findings that lung and ovarian cancer patients with *MEG3* over-expression have better response to cisplatin treatment[26–28]. Our model also identified previously reported regulation of cisplatin response by *HOTAIR*[29], *MALAT1*[30], and *NEAT1*[31]. Besides, we also uncovered novel interactions that potentially contribute to clinical outcome. For example, the expression of *LINC00992* in primary tumors increases along with the disease progression (Supplementary Fig. 2c) and correlates with poor patient survivals in multiple cancer types that are routinely treated with chemotherapy (Supplementary Fig. 5d). Meanwhile, *LINC00992* is identified as a drug-resistance predictor for many cytotoxic agents, including cisplatin (PS: 0.99) and gemcitabine (PS: 0.99). *LINC00992* overexpression-related chemo-resistance might account for the observed poor prognosis in patients with high *LINC0992* expression.

Notably, agents targeting the same pathway tended to share similar predictive lncRNAs (Fig. 2c, Supplementary Fig. 2b, d, Methods section). For example, agents targeting the genome integrity shared significantly more predictive lncRNAs ($p = 9.6 \times 10^{-9}$, Wilcoxon rank-sum test, Fig. 2f). Moreover, within the genome integrity group, PARP inhibitors olaparib and talazoparib shared a significantly higher proportion of predictive lncRNAs (OR = 3.32, $p = 1.6 \times 10^{-55}$, Fisher's exact test) than with CHEK inhibitor AZD7762 (OR = 1.44, $p = 8.9 \times 10^{-6}$, Fisher's exact test), indicating that lncRNA-drug predictive pairs might imply the mechanism that is involved in the drug response in these cell lines.

**LncRNA-based models predict drug response in cell lines**. Using the most predictive lncRNAs identified by the bootstrapping training, a lncRNA-based EN prediction model (LENP) was built for each agent (Supplementary Fig. 1a, Methods section). The model performance was assessed by tenfold cross-validation using Pearson's correlation coefficient and Kendall's $\tau$ of observed versus predicted IC50s (Supplementary Data 4, Methods section).

Here we refer to LENP models trained using IC50 values, but very similar results were obtained by using AUCs (Supplementary Fig. 3a). Compared to the previous bootstrapping procedure with all of the lncRNAs included, LENP models have a substantially improved performance in predicting the cell lines IC50s by using the top predictive lncRNAs (Fig. 3a). The improved model performance indicated the EN regression's power in identifying lncRNAs that are highly predictive of drug response. Overall, the pan-cancer LENP models reached a median performance at $r = 0.55$ ($p < 10^{-33}$, Pearson's correlation), while the cancer-specific LENP models have a median performance at $r = 0.71$ ($p < 10^{-6}$, Pearson's correlation) (Fig. 3b, see Methods section). Notably, agents with higher pan-cancer performance tend to be agents that

have a broader anti-cancer spectrum (Fig. 3c). For instance, in pan-cancer models, agents targeting the cell cycle, genome integrity and mitosis have overall better performances than agents targeting the ABL signaling and IGFR signaling (Fig. 3c). We also observed that some models built for targeted agents have increased performance in cancer-specific models compared to pan-cancer models. For example, the acute myeloid leukemia (LAML)-specific model for imatinib had an elevated performance ($r = 0.82$, Pearson's correlation) compared to the pan-cancer model in predicting the IC50s in leukemia cell lines ($r = -0.09$, Pearson's correlation).

Next, we sought to validate the LENP models using the CCLE and the CTRP databases (Supplementary Data 4, see Methods section). Among the 14 overlapped agents in CCLE database, LENP models successfully predicted the cell line response for 9 agents ($p < 0.05$, Spearman's correlation), including paclitaxel (rho = 0.34, $p = 0.0014$, Spearman's correlation) and 17-AAG (rho = 0.32, $p = 4.6 \times 10^{-7}$, Spearman's correlation) (Fig. 3d, e). For 76 overlapped agents in CTRP database, LENP models could predict the drug response for 34 of them ($p < 0.05$, Spearman's correlation). We observed a strong positive association between prediction performance and the inter-database drug sensitivity measurements consistency ($r = 0.59$, $p = 0.028$ for CCLE; $r = 0.62$, $p = 1.7 \times 10^{-9}$ for CTRP, Pearson's correlation), indicating the independent validation performance largely depends on the agreement between databases[32] (Supplementary Fig. 3b).

**LENP models predict therapeutic outcomes in cancer patients**. We have shown that cancer cell lines could recapitulate the lncRNA transcriptomic, genomic, and epigenetic alterations in primary tumors. To determine whether cancer cell line based LENP model could predict patient drug response, we applied the LENP models to TCGA tumor lncRNA expression profile and predicted patient drug response across 21 cancer types (Supplementary Figs. 1a, 4a, Supplementary Data 5, see Methods section). Since chemotherapy is widely used in advanced stages of solid tumors, the prediction is restricted to solid tumor patients with stage II (or later) disease and all LAML patients.

Our analysis revealed that LENP is capable of predicting the known and novel drug sensitivities in patients (Supplementary Fig. 4b, see Methods section). For example, bleomycin is an FDA approved agent to treat head-neck squamous cell carcinoma (HNSC), uterine corpus endometrial carcinoma (UCEC), cervical squamous cell carcinoma and endocervical adenocarcinoma (CESC). Compared to an average sensitivity rate (see Methods section) at 23.8% of other cancer types, a significantly higher sensitivity rate to bleomycin was observed in patients with UCEC (sensitive rate: 95.3%; $p = 3.59 \times 10^{-74}$, two-side KS test), CESC (sensitive rate: 55.5%; $p = 3.53 \times 10^{-16}$), and HNSC (sensitive rate: 38.5%; $p = 0.06$) (Fig. 4a, see Methods section). Besides these FDA approved indications (i.e. drug-cancer type pairs), our data suggest that 46 out of 49 (93.9%) drugs had a proportion of 'sensitive' tumors for which the treatment by such drugs has not been approved (Supplementary Fig. 4b, see Methods section). For

**Fig. 1** Cancer cell lines recapitulate the LncRNA alterations in primary tumors. **a** Genomic and epigenetic alterations of cancer-related lncRNAs in 505 cancer cell lines. Cell lines are arranged by columns. LncRNAs are arranged by rows. Three heatmaps indicate the patterns of the expression (left), DNA methylation (middle), and copy number (right) for cancer-related lncRNA (Methods section). Twenty-two cancer types are indicated by different colors on top of each heatmap. **b** Percentage of lncRNA genomic and epigenetic alterations occurring in at least one or at least three cell lines. **c-e** Pairwise Pearson's correlation of lncRNA alterations between cell lines and patient tumors for each cancer-type in CNV, methylation, and expression. The correlation of lncRNA alteration within the same and different cancer types are shown in the boxplots (center lines represent median correlation, the box limit indicates the lower quantile and upper quantile, and whiskers represent the minimal and maximal correlations). **f** Performance of the nearest-neighbor matching algorithm to predict cell origin using expression, methylation, CNV and merged features, respectively

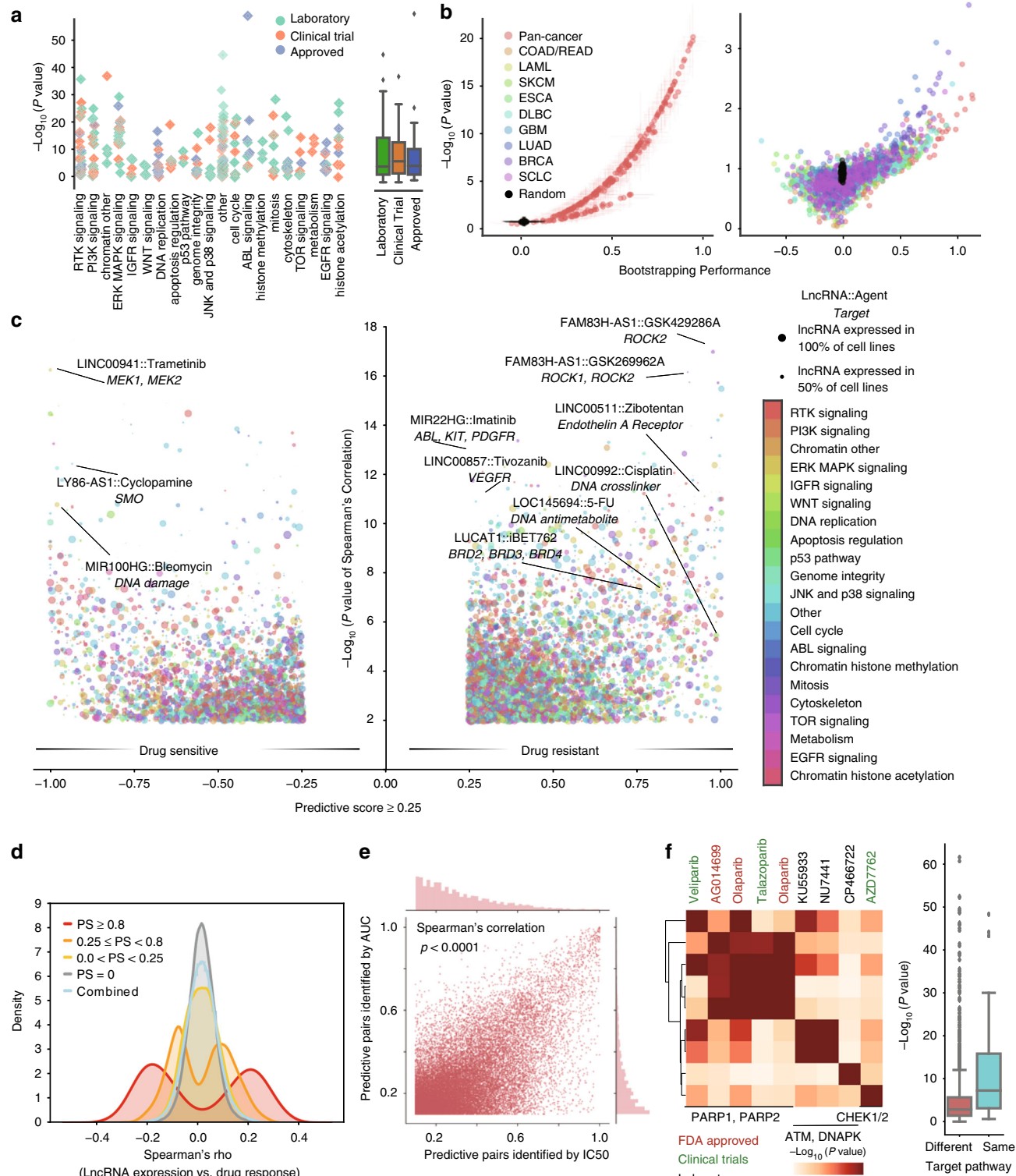

**Fig. 2** The landscape of LncRNA-drug predictive pairs in cancer cell lines. **a** Effect of cell lineage on drug response prediction for each agent. The linage effect is evaluated using one-way ANOVA, and negative log10-transformed *p*-values are indicated on *y*-axis. The agents are organized based on targeting pathways (*x*-axis). **b** Volcano plot of pan-cancer models (left) and cancer-specific models (right) performance in drug response prediction in the bootstrapping process. Pearson correlation coefficients (*x*-axis) and negative log10-transformed *p*-values (*y*-axis) indicate the model performance. **c** lncRNA-drug predictive pairs landscape across 265 agents and 505 cancer cell lines. The predictive score for each lncRNA-drug interaction and the negative log-transformed *p*-value for Pearson's correlation between the lncRNA expression and IC50 were shown in the *y*-axis and *x*-axis of the volcano plot. **d** The distribution of Spearman's correlation coefficients between lncRNA expression and ln-transformed IC50s. The density plot of the coefficients is shown for (i) strong predictive pairs with PS >= 0.8; (ii) moderate predictive pairs with 0.25 <= PS < 0.8; (iii) weak predictive pairs with 0 <= PS < 0.25; (iv) non-predictive pairs and (v) the combination of all lncRNA-drug pairs. **e** Scatter plot of the predictive score between lncRNA-drug predictive pairs identified in IC50 models and those identified in AUC models. **f** Agents targeting genome integrity clustered by shared predictive lncRNA signatures. One-sided Fisher exact test *p*-values were indicated by different colors in the heatmap

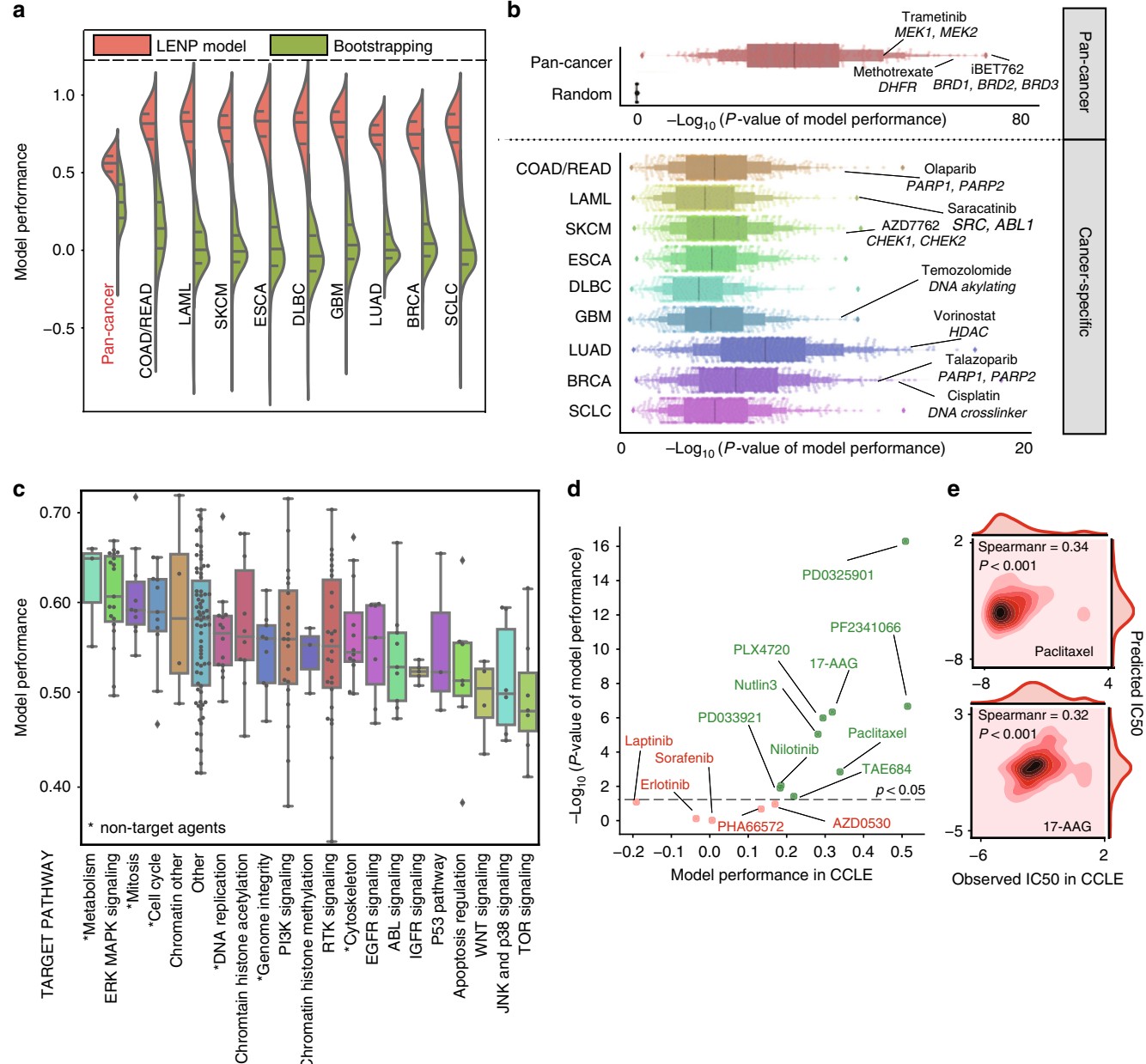

**Fig. 3** LncRNA-based EN-prediction models predict drug response in cancer cell lines. **a** Performance comparison between LENP and bootstrapping EN models for 265 drugs in pan-cancer and specific cancer types. Model performance is shown on the y-axis. **b** LENP performance of pan-cancer models and cancer-specific models using top 20 predictive lncRNAs for each agent. **c** Pan-cancer LENP performance for agents from different target pathways. center lines represent median performance, the box limit indicates the lower quantile and upper quantile, and whiskers represent the poorest and best performance. **d** Prediction performance of LENP models in CCLE data. The performance is assessed by Spearman correlation coefficients (x-axis) and −log10-transformed p-value of real IC50s in CCLE versus predicted IC50s by lncRNA-based EN models. Label colors demonstrated the significance: the model with p-value <0.05 is considered as having good independent validation performance. **e** Performance of LENP in selected agents from CCLE datasets. Observed and predicted IC50 for CCLE dataset were represented in the x-axis and y-axis, respectively

example, Imatinib is an FDA approved agent to treat chronic myeloid leukemia (CML). Based on LENP model, 100% of acute myeloid leukemia (AML) patients are predicted to be imatinib sensitive ($p = 2 \times 10^{-15}$, two-side KS test) (Fig. 4a). Besides, ~74.2% of patients with glioblastoma (GBM) ($p = 7.41 \times 10^{-6}$, two-side KS test) and 99.1% of patients with low-grade glioma (LGG) ($p = 3.96 \times 10^{-60}$) were predicted to be sensitive to imatinib (Fig. 4a). Although this drug is not currently approved for glioblastoma or AML treatment, phase II clinical trials have been carried out to test the efficacy of imatinib in treating glioblastoma and AML[33–35].

Because the TCGA cancer patients were mostly treated based on standard chemotherapy protocol[36], we hypothesized that patients would likely to undergo poor prognosis if they were predicted to be resistant to therapies. For 49 FDA approved therapeutic agents, we observed 66 significant associations between predicted drug response and poor survival outcome in different cancer types ($p \leq 0.05$, univariate Cox regression) (Fig. 4b, see Methods section).

In clinic, patients usually take a combination of different drugs rather than single agents. Thus, to better study the chemotherapy response of cancer patients, we give each patient a consensus drug

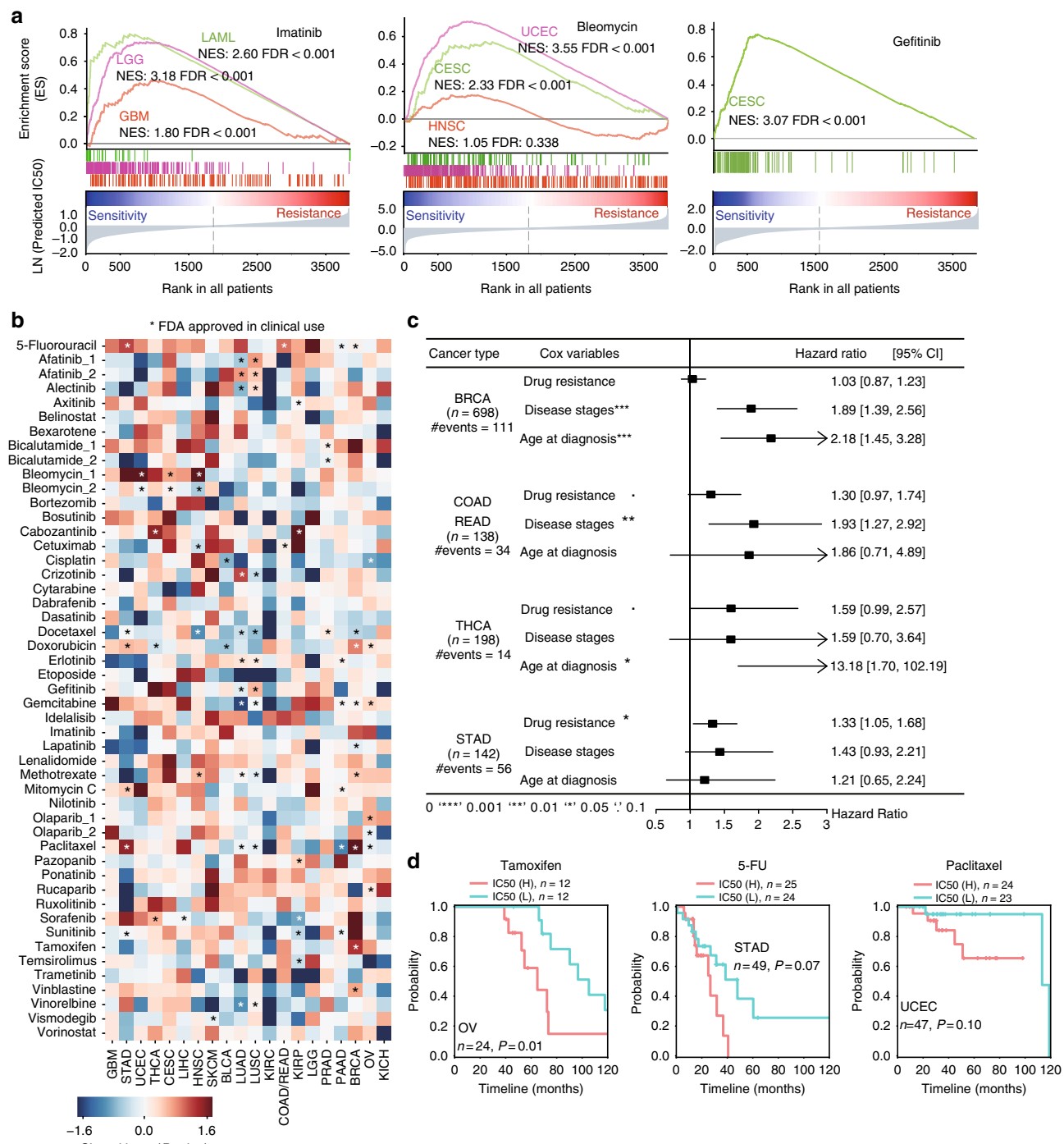

**Fig. 4** LncRNA-based EN-prediction models predict drug response in patient tumors. **a** Imatinib (left), bleomycin (middle), and gefitinib (right) were predicted to be sensitive in FDA approved or yet-to-be approved cancer types. **b** Different overall survival rates between patients with different predicted responses to each of the 49 FDA approved cancer drugs in individual cancer type. Blue and red represented hazard ratio by the univariate cox regression model. The asterisks indicate the drugs that are approved by FDA for corresponding cancer types. **c** Forest plot of multivariate Cox regression analysis of "Drug resistance", "Stage" and "Age at diagnosis" on patient survival in four cancer types. **d** The Kaplan–Meier curves of overall survival for OV, STAD, and UCEC patients who received corresponding treatments. The patients were segregated by median predicted IC50s

response score by combining the prediction of first- and second-line chemotherapy (Supplementary Data 5) that are approved by FDA for each cancer type (see Methods section). Using this heuristic method, we observed a trend in poor prognosis among patients that are predicted to be chemotherapy resistant in THCA (Thyroid Carcinoma, hazard ratio (HR) = 1.76, $p = 0.05$, multivariate Cox regression), STAD (Stomach Adenocarcinoma, HR

= 1.40, $p = 0.02$) and CRC (Colorectal Cancer, HR = 1.38, $p = 0.08$) after adjusting known prognostic factors such as age at diagnosis and disease stages (Fig. 4c, Supplementary Fig. 4c, d, see Methods section).

To further test the LENP model in the patients that actually received the corresponding chemotherapy, we parsed the chemotherapy treatment data from TCGA patient clinical

information (see Methods section). Although most cancer patients' chemotherapy treatment data are missing, several cancer types, including breast cancer, ovarian cancer, uterine, and gastric cancer, have relatively complete chemotherapy treatment history in record. We found LENP can predict the therapeutic outcome for a number of drugs. For example, there are 24 ovarian cancer patients and 210 breast cancer patients received tamoxifen treatment in TCGA cohorts. Among those patients, poor survival is observed for patients with predicted drug resistance by LENP-tamoxifen model (log-rank test: $p = 0.01$, OV; $p = 0.19$, BRCA) (Fig. 4d, Supplementary Fig. 4c, see Methods section). When applying LENP-paclitaxel model to 111 breast cancer patients, 138 ovarian cancer patients and 47 endometrial cancer patients who have been treated with paclitaxel, we also observed a trend of poorer survival among patients with predicted drug resistance (log-rank test: $p = 0.12$, BRCA; $p = 0.30$, OV; $p = 0.10$, UCEC) (Fig. 4d, Supplementary Fig. 4c, see Methods section). In addition, we applied LENP-5FU model to 49 fluorouracil-treated stomach adenocarcinoma patients. We found that patients with lower IC50s predicted by LENP-5FU model tend to have a better survival outcome compared to the rest (log-rank test: $p = 0.08$, STAD) (Fig. 4d, see Methods section).

In addition, we also constructed the protein-coding gene (PCG) based models for 49 FDA approved agents using the similar training-testing framework. We compared the performance of PCG and LENP models in predicting cell line response and the patient survival outcome. For drug response prediction in both cell lines and patients, we observed overall comparable performances between LENPs and PCGs (Supplementary Fig. 4f). Interestingly, the LENP models outperform PCG-based models in many cases including the aforementioned ones. For example, LENP-tamoxifen model could better predict the prognosis of OV patients treated with tamoxifen (log-rank test: $p = 0.01$; HR = 3.62; 95% confidence interval (95%CI), 1.26–11.13) than PCG-based model (log-rank test: $p = 0.13$; HR = 2.32; 95% CI, 0.74–7.24). The predicted 5FU resistance by LENP model is better associated with poor survival of STAD patients (log-rank test: $p = 0.07$; HR = 2.24; 95% CI, 0.89–5.64) than that of PCG-based model (log-rank test: $p = 0.66$; HR = 1.23; 95% CI, 0.49–3.09). In the cases of paclitaxel-treated OV and UCEC patients, the LENP models predicted resistant patients would undergo poor prognosis; however, the PCG-based models predicted resistant patients would undergo a beneficial prognosis (Supplementary Fig. 4g).

**LncRNA may regulate drug resistance through drug metabolism.** LncRNAs have been reported to regulate the cancer drug response through regulating the protein-coding genes involved in drug-metabolism and drug-target pathways[19,20]. Since multi-drug resistance remains a major obstacle of successful chemotherapy in clinical treatment of primary and recurrent disease[37], we are particular interested in lncRNAs that are predictive of multi-drug response of agents with different mechanisms (Fig. 5a, Supplementary Fig. 2h). By using an entropy-based approach, we identified 381 potential multi-drug response (MDR) related lncRNAs that are independent from drug-target mechanism (Supplementary Data 6, see Methods section). To determine the possible functional roles of these lncRNAs, we performed Gene Sets Enrichment Analysis (GSEA)[38] on the co-expression profile between lncRNAs and protein coding genes (Supplementary Fig. 5a; see Methods section). We observed a significant correlation between MDR lncRNAs and xenobiotic metabolism (rho = 0.32, $p = 5.76 \times 10^{-55}$, Spearman's correlation), glycolysis (rho = 0.29, $p = 7.13 \times 10^{-47}$), apoptosis related pathways (rho = 0.34, $p = 4.11 \times 10^{-57}$, Fig. 5b, Supplementary Fig. 5b, Supplementary

Data 6) and ABC transporters (rho = 0.19, $p = 2.01 \times 10^{-16}$, Spearman's correlation). Previous studies have highlighted the remarkable contribution of xenobiotic metabolism, glycolysis and apoptosis in inducing multi-drug resistance[39,40]. Genes involved in xenobiotic metabolism (e.g., cytochrome P450 genes) could regulate the drug metabolism and modulate the intracellular drug concentration, which would result in drug resistance and heterogeneous response among individual tumors[37,39,40].

Our analysis identified 164 MDR-related lncRNAs that are significantly correlated with xenobiotic metabolism (FDR < 0.25, GSEA) (Fig. 5c, Supplementary Data 7). LINC00992 (a.k.a. CTC-504A5.1) is identified as one of these MDR lncRNAs. LINC00992 is an intergenic lncRNA located on chromosome 5q23.1 and is expressed in multiple cancer types (Supplementary Fig. 5d). Being predictive of cell line response of 118 agents (Fig. 5e, Supplementary Data 7), LINC00992 exhibited significant positive expression correlation with CYP2J2 (r = 0.29, Pearson's correlation, $p < 0.001$), CYP1A1 (r = 0.21, Pearson's correlation, $p < 0.001$) as well as several other genes involved in xenobiotic metabolism pathway (NES: 1.25, FDR < 0.01, GSEA) (Fig. 5d, Supplementary Fig. 5b, c). Cancer cell lines with high expression of LINC00992 and CYP genes showed resistance to 116 (98.3%) of the predictable agents (Fig. 5e–g). Furthermore, elevated expression of LINC00992 associated with poor survival in patients of BRCA ($p = 0.022$, log-rank test), LIHC ($p = 0.065$), THCA ($p = 0.024$) and READ ($p = 0.178$) (Fig. 5h, i, Supplementary Fig. 5e and f). Interestingly, LINC00992 has been identified as a potential regulator of CYP genes[41], which play important roles in chemotherapy resistance in cancer[37,39,40]. Therefore, LINC00992 may serve as a novel biomarker and a potential master regulator for multi-drug resistance through xenobiotic metabolism.

**EPIC1 as a master regulator of BET inhibitor resistance.** In addition to the drug metabolism pathways, our analysis also revealed lncRNAs that regulate the drug response directly through drug-target pathways (Supplementary Fig. 6a, Supplementary Data 8, see Methods section). For example, estrogen response pathway significantly correlated with expression of 14 out of 20 (70%) top predictive lncRNAs in the pan-cancer tamoxifen EN-model (Fig. 6a). The top predictive lncRNAs for PARP1/2 inhibitor, including olaparib (FDA approved) and talazoparib (in clinical trial), demonstrated significant co-expression with genes in DNA repair (85% of top predictive lncRNAs for olaparib; 70% for talazoparib) and G2M checkpoint (85% for olaparib and 70% for talazoparib) (Fig. 6a). Intriguingly, top predictive lncRNAs of Bromodomain and Extra-Terminal inhibitors (iBETs) are significantly correlated with MYC-related pathways (80% for iBET762 and 85% for JQ-1) (Fig. 6a). This is consistent with the previous reports that iBETs achieve therapeutic effect in multiple cancer types by targeting c-MYC pathway[42-48].

The iBETs are a class of small molecules that could block the function of BET protein family. The iBETs have been demonstrated to be a promising new therapy in several cancer types, including breast cancer[46,49]. In our study, both pan-cancer and BRCA-specific LENP models can predict iBETs drug response with high sensitivity and specificity (Fig. 6b, Supplementary Fig. 6b). Among the novel predictive features for BET inhibitors responses are RP11-275I4.4 and RP11-708B6.2 (top predictors of iBETs sensitivity), as well as EPIC1 (top predictor of iBETs resistance, Fig. 6c).

EPIC1 (Epigenetically induced lncRNA 1)[24] is an intergenic lncRNA located on chromosome 22q13.31 and is highly overexpressed in 15 cancer types including BRCA (Supplementary Fig. 6c; Supplementary Data 9). Being selected as a top

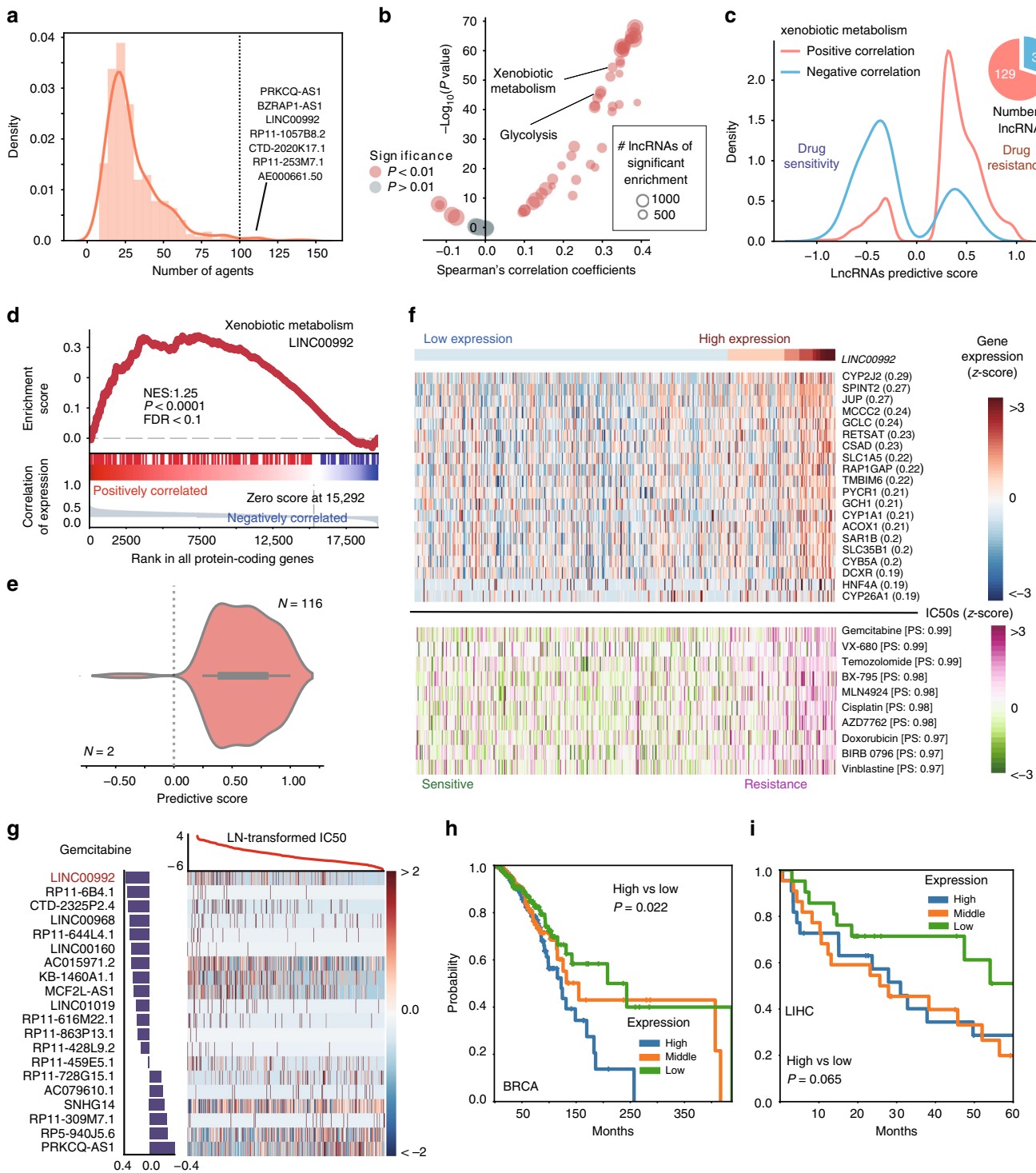

**Fig. 5** Co-expressed pathways reveal that LncRNAs induce multi-drug resistance by regulating xenobiotic metabolism pathway. **a** Distribution of the number of agents that multi-agent response (MDR) related lncRNAs are predictive to. The listed are the top five lncRNAs predicting the greatest number of agents' response in cell lines. **b** Correlation between the Shannon entropy of lncRNAs and their absolute normalized enrichment score across cancer hallmarks pathways. **c** Marginal distribution of predictive score in pan-cancer models. The red (blue) color denotes MDR-related lncRNAs that are positively (negatively) associate with xenobiotic metabolism genes. The pie chart on the right indicates the ratio between two groups of lncRNAs. **d** GSEA analysis of association between *LINC00992* and xenobiotic metabolism genes. **e** Distribution of *LINC00992*'s predictive score across the agents. **f** The association among the high expression of *LINC00992*, genes in xenobiotic metabolism and the IC50s of top predictable agents across cancer cell lines. The upper heatmap shows the expression level from blue (low) to red (high) colors. The lower heatmap shows the IC50s from green (sensitive) to purple (resistant) colors. **g** Pan-cancer LENP model for gemcitabine. The top curve shows observed IC50 of gemcitabine in each cell line. The central heatmap shows the top predictive lncRNA expression in the model across all cell lines (*x*-axis). Bar plot (left): weight of the top predictive lncRNAs in the model for gemcitabine sensitivity (bottom) or insensitivity (top). **h**, **i** The Kaplan–Meier curves of overall survival for patients grouped into three groups of equal sample size, i.e., high, median, low, by *LINC00992* expression level in BRCA and LIHC

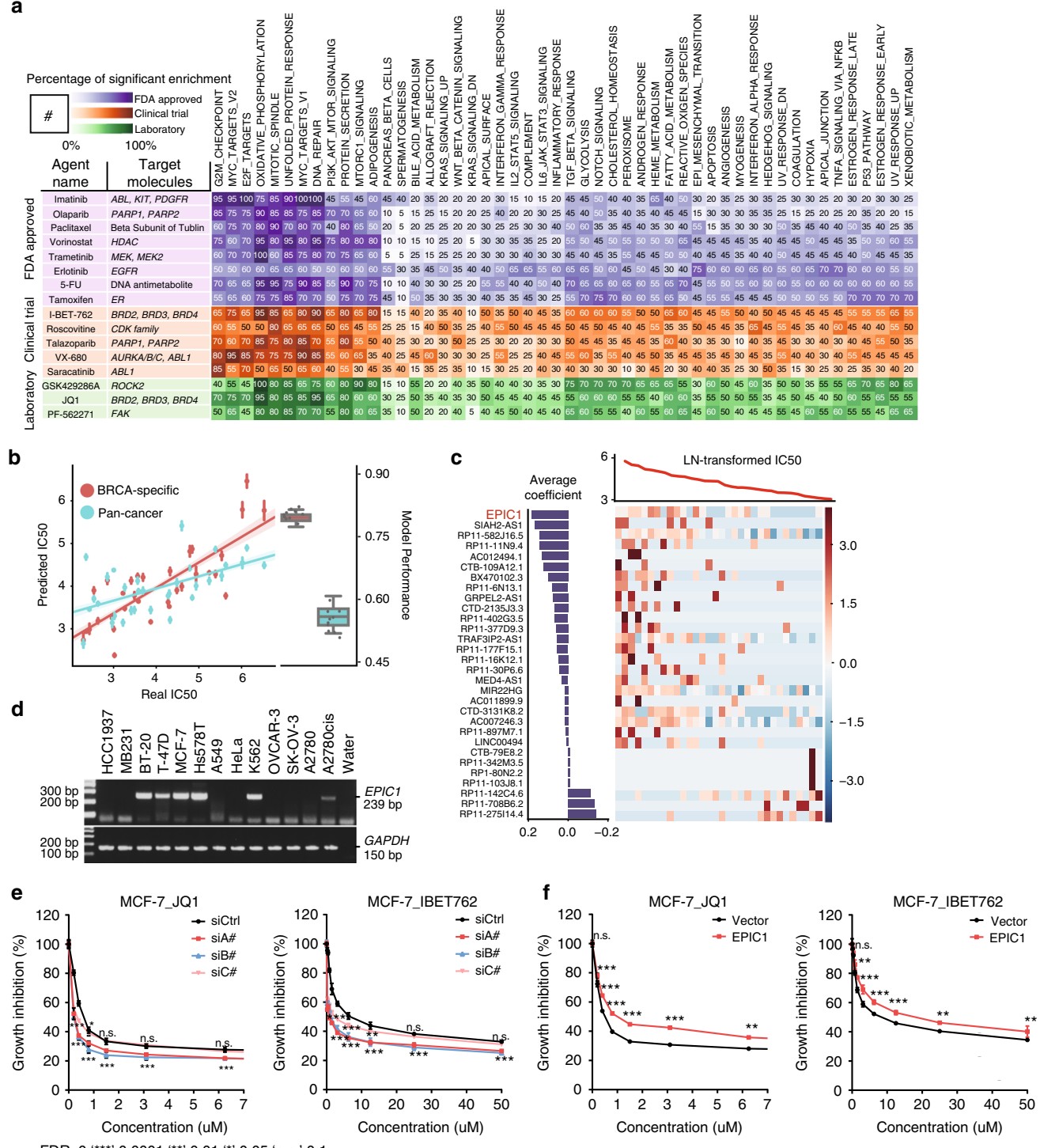

**Fig. 6** *EPIC1* overexpression enhances breast cancer cell lines resistance to BET inhibitors. **a** Enrichment of top predictive lncRNAs for each agent in cancer hallmark pathways. The left panel lists the target information of the agents. The right panel shows the number of predictive lncRNAs that are significantly associated with cancer hallmarks (FDR < 0.25 by GSEA). **b** Comparison of LENP model predicted IC50 in ten iterations and observed IC50 for I-BET-762. Model performance in ten iterations for both pan-cancer and BRCA-specific models were demonstrated in the box plot. center lines represent median performance, the box limit indicates the lower quantile and upper quantile, and whiskers represent the poorest and best performance. Data are presented as mean ± SE (*n* = 10 for 10 iterations of CV). **c** LENP model for I-BET-762 in BRCA. The top curve shows observed IC50 of I-BET-762 in each cell line. The central heatmap shows the top predictive lncRNA expression in the model across all cell lines (*x*-axis). Bar plot (left): weight of the top predictive lncRNAs in the model for I-BET-762 sensitivity (bottom) or insensitivity (top). **d** Endogenous expression level of *EPIC1* in 13 cell lines and water. **e** Growth inhibition curves for *EPIC1* knockdown or control MCF-7 cells treated with BET inhibitor I-BET-762 (left) and JQ-1 (right). **f** Growth inhibition curves for *EPIC1* overexpression or control MCF-7 cells treated with BET inhibitor I-BET-762 (left) and JQ-1 (right). Data are presented as mean ± SD (*n* = 3 for technical replicates)

predictor of iBET resistance by BRCA-specific LENP-iBET model (Supplementary Data 9), *EPIC1* expression has a significant positive correlation with IC50s of iBET762 in breast cancer cell lines (rho = 0.53, $p$ = 0.002, Spearman's correlation) (Supplementary Fig. 6d). Moreover, high expression of EPIC1 is also associated with poor survival in patients of BRCA ($p$ = 0.067, univariate Cox regression), UCEC ($p$ = 0.014), KIRC ($p$ = 0.0004) and COAD ($p$ = 0.052) (Supplementary Fig. 6e).

We next designed primers to screen *EPIC1*'s expression in 13 cell lines using RT-PCR (see Methods section). *EPIC1* is upregulated in MCF-7, ZR-75-1, and Hs578T cell lines and is expressed at low levels in A549 cell line (Fig. 6d, Supplementary Fig. 7a). To investigate the *EPIC1*'s role in regulating the iBET response in cancer cell lines, we knocked down the *EPIC1* expression in MCF-7, BT-474 and ZR-75-1 breast cancer cell lines with three *EPIC1* siRNAs. Knockdown of *EPIC1* significantly increased the iBETs sensitivity in MCF-7, BT-474, and ZR-75-1 cells (Fig. 6e, Supplementary Fig. 7d). We further cloned the full-length human *EPIC1* cDNA[24] and overexpressed *EPIC1* in MCF-7 breast cancer cells and A549 lung cancer cells. In accordance with our LENP prediction, overexpression of *EPIC1* led to the drug resistance of iBET in MCF-7 and A549 cells (Fig. 6f, Supplementary Fig. 7e).

To further explore the underlying mechanism of *EPIC1* in regulating iBETs resistance, RNA-seq analyses were performed in four cancer cell lines including MCF-7 and Hs578T cells after *EPIC1* knockdown by *EPIC1* siRNAs, individually or pooled (GEO: GSE98538). We focused only on genes regulated in the same direction in all three transfections to exclude the possible siRNA off-target effects. *EPIC1* knockdown in breast and ovarian cancer cells resulted in significant expression change of 4318 genes, which were significantly overlapped with *EPIC1*-correlated genes in 505 cancer cell lines ($p$ < 0.0001, two-side Fisher's exact test) (Supplementary Fig. 8a, see Methods section). Moreover, 16 out of 18 *EPIC1*-correlated pathways in 505 cancer cell lines are significantly regulated by *EPIC1*-knockdown (FDR < 0.25, GSEA) (Supplementary Fig. 8b, see Methods section). Among them, the MYC pathway/targets are prominent gene sets enriched with *EPIC1*-associated genes in both cancer cell lines and *EPIC1*-knockdown cells (Supplementary Fig. 8c, d). In another study of our group[24], we have mechanistically demonstrated that *EPIC1* regulates MYC transcriptional activity by directly interacting with MYC protein. Overexpression of *EPIC1* increased MYC target expression and breast tumorigenesis in vitro and in vivo, which can be abolished by *MYC* knockdown[24]. Our observations suggest that *EPIC1* is an oncogenic lncRNA and also plays an important role in promoting the resistance to iBETs by increasing MYC protein's transcriptional activity.

## Discussion

The study of lncRNAs' role in cancer drug response has not gained much momentum due to the dearth of genomics/epigenetic platforms covering the non-coding region of the human genome and the paucity of information regarding drug response in tumors. These bottlenecks have led the majority of lncRNA studies to use a "bottom-up" strategy by first determining each individual lncRNA's downstream regulatory function and then investigating the lncRNA's regulation of drug response in cancer. In this project, we have adopted a "top-down" approach, which integrates the pharmacogenomics data from both primary tumor and cancer cells to construct the lncRNA-based drug response prediction models and to identify the candidate lncRNAs that may mechanistically regulate drug response. We have shown that cancer cell lines could highly recapitulate the lncRNAs alterations in primary tumors. Moreover, cancer cell line based EN-models,

i.e., LENP models, could readily predict chemotherapy responses in patients with breast, stomach, ovarian, and endometrial cancer. The integrative analyses between lncRNA and protein-coding gene expression further suggest that lncRNAs might regulate cancer drug response through regulating drug-metabolism or drug-target pathways. To our best knowledge, this is the first study to systematically link non-coding genotypes with drug response phenotypes in both cancer cell lines and patient tumors.

By further applying the LENP model to cancer patients who have actually received the corresponding chemotherapy, we have shown that LENP can readily predict those patients' chemotherapy response. Interestingly, lncRNA-based models outperformed protein-coding models in those cases. Although we were only able to validate the cell line based LENP models in 4 cancer types and three drugs due to the complexity of chemotherapy that was given to each individual cancer patient, these serve as proof of principles for using the non-coding genotype in cell-line-based panels to gain insights into precision cancer medicine. With the emerging of the pharmacogenomics data of standardly designed cancer precision medicine project like GENIE[50], we should be able to determine the performance of lncRNA-based EN-models in patient tumor in short future.

One drawback of the lncRNA-based EN-models in present study is that we only include the lncRNA expression levels to train the EN-models, because (1) the lncRNA expression exhibits the highest similarity between cancer cell lines and patient tumors; (2) the changes of both CNA and DNA methylation will eventually be manifested by the expression of lncRNA, and (3) the redundancy of including lncRNA CNA and DNA methylation data may not be properly handled by the EN-model in current study. Moreover, we didn't include the PCGs and other non-coding genes such as miRNAs into the model for drug prediction. Comparing to the lncRNAs, secreted proteins and miRNAs have been well documented to be premium candidates for biomarkers given their higher endogenous expression levels and better stability in detection[51,52]. Emerging deep-learning algorithms, such as artificial neural networks, have shed light to modeling high-dimension and high-redundancy data[53]. In future study, we will use deep-learning algorithm to comprehensively model the cancer drug response by integrating lncRNA, miRNA, and PCG genomics and epigenetic changes.

On top of drug prediction models, our feature selection strategy has also identified 27,341 lncRNA-drug predictive pairs, which may help reveal novel lncRNA regulators for drug response in cancer. However, our ability to validate these noncoding pharmacogenomic associations in independent databases (i.e., CCLE and CTRP) was largely restricted by (1) the use of different pharmacological assays, (2) the limited number of overlapping cell lines and agents, and (3) the lack of gold standard to define "true positives" among those pharmacogenomic databases[32,54]. We believe the ongoing efforts in standardizing and optimizing the drug-response measurements will greatly reduce the inconsistency and improve the robustness of pharmacogenomic associations across different studies in the short future[22,32]. Therefore, the lncRNA-drug predictive pairs in our study still require experimental validation and mechanism investigation. In this regard, we have experimentally demonstrated that *EPIC1*, the top predictive lncRNA for iBET drug response, regulates iBET resistance in breast cancer by regulating MYC transcriptional activity.

The iBETs are a class of MYC inhibitors, which have been demonstrated to have great potential to be translated to clinic in several cancer types, including breast cancer[46,49]. The success of targeting MYC by iBET[46,55,56], with only minor toxicity in patients[57], has potentiated iBETs as a very promising class of agents for cancer therapy. However, the resistance to iBET, which

was recently reported in multiple cancer types such as leukemia and breast cancer, has largely hindered their translation into clinic[47,49,58]. Despite that tremendous effort has been invested to identify the underlying regulator and biomarker for iBETs resistance, the detailed mechanism remains elusive. In our recent publication[24], we have revealed that *EPIC1* can be epigenetically activated by loss-of-DNA methylation at its promoter region in multiple cancer types, including breast cancer. We have shown *EPIC1* can directly interact with MYC protein and regulate its transcriptional activity. Our results suggest that *EPIC1* may regulate iBET resistance through increasing MYC protein's transcriptional activity. Future mechanistic study is warranted to test this hypothesis.

In summary, the current study established a detailed knowledge base of lncRNA-drug predictive pairs in 27 cancer types and 265 agents. This should facilitate the ongoing effort in developing novel genomic models for cancer precision medicine. Moreover, the further identification of potential lncRNA master regulators of cancer drug response will pave the way for lncRNA-based cancer therapy in the short future by providing promising therapeutic targets to overcome the cancer chemotherapy resistance.

## Methods

**Pre-processing of lncRNAs alteration data.** For cancer cell lines, expression of 13,335 lncRNAs across 505 cancer cell lines from Cancer Cell Line Encyclopedia (CCLE) was downloaded from Expression Atlas[59] with matched drug response data from Genomics of Drug Sensitivity in Cancer (GDSC). For patient samples, expression of 2614 cancer-related lncRNAs in 5605 TCGA patient samples was downloaded from MiTranscriptome[11]. Expression levels of these lncRNAs are logarithmic transformed and z-score normalized for both cell lines and patients. We obtained the lncRNAs copy number alteration data for both 505 cell lines and 5605 TCGA patient samples by mapping 12,139 Affymetrix SNP 6.0 microarray segmentations to 2614 lncRNA regions. For DNA methylation, we repurposed Illumina 450K Human Methylation microarray to get beta values of 2804 unique probes for lncRNAs in (i) 1028 cell lines from COSMIC[22] and (ii) 5605 patients from TCGA.

**Pre-processing of drug response data.** Drug response data of 265 agents across 1001 cancer cell lines were downloaded from GDSC database[22]. These 265 agents include 49 clinical drugs, 84 drugs in clinical development and 132 experimental agents. The drug response in each cell line is indicated by logarithmic transformed IC50s and AUCs. 505 cell lines with genomic data available are retained for model training and following analysis. Clinical drug treatment information of 10,237 TCGA patients across 33 cancer types were parsed from the National Cancer Institute (NCI) Genomic Data Commons (GDC) using the R functions 'GDCquery', 'GDCDownload' and 'GDCprepare_clinic' provided by TCGAbiolinks[60].

**Cell lines versus tumors comparisons.** The comparison between cell lines and tumors was based on feature correlations and nearest-neighbor matching with the average, broken down by different cancer types (Fig. 1c–f). A bootstrapping procedure was performed for each comparison: for each cancer type, we calculate the fold-change for each genomic feature (e.g. gene expression, methylation, copy number alterations) between that cancer type and a resampling of all other cancer types. To ensure the representation of homogeneous tissue-type, we only retained cancer types with primary tumor samples more than 15 and cell line samples more than 20. Next, we calculated the pairwise Pearson's correlation coefficient of the fold-changes between cell lines and primary tumors. This procedure would be iterated for 10 times with different samplings. The final asymmetric correlation matrix for each genomic feature is an average matrix of coefficients obtained by 10 iterations. The diagonal demonstrated the agreement between cell lines and tumors within the same cancer type. For the analysis merging three genomic features together, the integrated correlation matrix is calculated based on the rank sum of each feature in the original correlation matrix. A comparison with p-value fell into the first 10% percentile would be considered as significant correlation (Fig. 1c–e). This correlation matrix was further used to fit the nearest-neighbor matching algorithm[22] (Fig. 1f). This algorithm will determine if the input cancer type of patients could correctly match the cancer type of cell lines within the top $k$ ($k = \{1, 2, 3, 4, 5, 6, 7\}$) nearest neighbors. A percentage of top k nearest neighbors that contain a correct match is calculated for matching evaluation.

**Identification of lncRNA-drug predictive pairs.** To identify lncRNAs that were most associated with drug response, we applied Elastic Net regression[61,62], a machine learning approach, combined with a bootstrapping procedure for each of the 265 agents. For each compound, this algorithm would pick up group of lncRNAs whose expression pattern could best explain the drug sensitivity profiles of the cell lines. The Elastic Net regression is a well-demonstrated model to work with the conditions in which the number of features is far greater than the number of observations. Before our study, many high-profile studies have already applied this regression algorithm to identify critical genomic features that could predict drug response in cancer cell lines[2,22].

For each compound, we constructed a drug response vector $Y \in R^{N,1}$, where $N$ is the number of cell lines treated with this compound. The values in the vector represent the drug responses across these cell lines, i.e., logarithmic transformed IC50 or area under the curve (AUC). For these cell lines, we then constructed an lncRNA expression matrix $X \in E^{N,1}$, where $N$ is the number of cell lines and $p$ is the number of lncRNAs. With input of $Y$ and $X$, we then used the scikit-learn 0.17.0 package [4] to solve the following optimization problem:

$$\min_{(\beta_0,\beta)\in R^{p+1}} \operatorname{Reg}_\lambda(\beta_0,\beta) = \min_{(\beta_0,\beta)\in R^{p+1}} \left[ \frac{1}{2N}\sum_{i=1}^{N}(y_i - \beta_0 - x_i^{\mathrm{T}}\cdot\beta)^2 + \lambda P_\alpha(\beta) \right],$$

where

$$P_\alpha(\beta) = (1-\alpha)\frac{1}{2}\parallel \beta \parallel_{L_2}^2 + \alpha \parallel \beta \parallel_{L_1}.$$

In this equation, $\alpha$ controls the ratio of the L1 and L2 penalty terms, while $\lambda$ controls the overall weight of the regression penalty. The optimization begins with 10 values of $\alpha \in [0.2,1.0]$ and 200 values of $\lambda = e^\tau$ with $\tau \in [-5,5]$. The optimal $\alpha$ and $\lambda$ that lead to the minimal mean square error of the regression model is obtained by tenfold cross-validation.

Next, we implemented a bootstrapping strategy to identify the most predictive lncRNAs for each compound. This procedure would generate 200 resampled datasets with replacement from the complete sample sets, $(X^{BS_i}, Y^{BS_i})_{i=1,2,\ldots,200}$, where $X^{BS_i} \in E^{N,p}$ and $Y^{BS_i} \in R^{N,1}$. Based on the optimal $\alpha$ and $\lambda$, the elastic net equation would be solved for each of the resampled datasets, and a regression coefficient matrix $\beta^{BS}$ would finally be built for each compound (Fig. 2b, c and Supplementary Fig. 1b).

To assess the extent to which an lncRNA is associated with the drug response, we then created a metric named 'predictive score' based on how frequent this lncRNA is selected by the regression model during the bootstrapping. For each lncRNA $u$ of each compound, $u \in \{1,2,\ldots,p\}$, the predictive score of lncRNA $u$ is then calculated by:

$$PS_u = \frac{1}{200}\sum_{j=1}^{200} I\left(\beta_{u,j}^{BS}\right),$$

where

$$I(x) = \begin{cases} 0, x = 0 \\ 1, x \neq 0 \end{cases}.$$

We have included one additional independent dataset Cancer Therapeutics Response Portal (CTRP)[23] to determine the specificity of this analysis under different stringencies. The specificity is defined as the distribution distance between the correlation of lncRNA expression and drug response in predictive or non-predictive lncRNA-drug pairs in the independent datasets. As shown in Supplementary Fig. 2a, the specificity of the predictive lncRNA-drug pairs increases as the PS cutoff getting more stringent. The specificity of the predictive lncRNA-drug pairs reaches to a turning point at PS cutoff = 0.25. We thus set 0.25 as cutoff to define lncRNA-drug predictive pairs. (Fig. 2c and Supplementary Fig. 2g). The same feature selection procedure has also been applied to the overlapped cell lines and agents in CTRP and CCLE to determine the robustness of the above strategy.

**Pairwise comparison of feature selection.** To compare the similarity of predictive lncRNA sets between agents, we used three different measurements to perform the pairwise comparison: Fisher's exact test. For lncRNA set of each compound $d$ ($d \in \{1,2,\ldots,26\}$), we dichotomized their predictive scores to 0 and 1 based on whether an lncRNA is considered as predictive or not. This operation would generate a binary vector $B_d^{p,1}$ for each compound $d$, where $p$ is the number of lncRNAs. Next, a similarity score matrix would be built based on the pairwise comparison of $\left(B_d^{p,1}\right)_{d \in \{1,2,\ldots,265\}}$ by performing Fisher's exact test. The resulting matrix would then be analyzed by hierarchical clustering using 'average' linkage and Euclidean distance (Fig. 2d and Supplementary Fig. 2h).

**Lineage effect on drug response and lncRNA expression.** ANOVA are used to evaluate the contribution of lineages to the drug response. For ANOVA, the cell lines are grouped by cancer types, following by the comparison between the inter- and intra-type variance of drug responses for each compound. A significant p-value indicates that the response of that drug is likely to be cancer specific (Fig. 2a).

**Machine learning models for drug response**. For each of the 265 agents, we selected top 20 lncRNAs with highest predictive score to build predictive models of drug response with Elastic Net regression. For each compound, we constructed a drug response vector $Y \in R^{N,1}$, where $N$ is the number of cell lines treated with this compound. The values in the vector represent the drug responses across these cell lines, i.e. logarithmic transformed IC50 or area under the curve (AUC). For these cell lines, we then constructed an lncRNA expression matrix $X \in E^{N,20}$, where $N$ is the number of cell lines. With input of $Y$ and $X$, we optimize the parameters with 10 values of $\alpha \in [0.2, 1.0]$ and 200 values of $\lambda = e^{\tau}$ with $\tau \in [-5,5]$ by tenfold cross-validation. Using optimal parameters, we build the final model $Y = f(X)$ for each compound and estimate the predictive power by 10 iterations of tenfold cross validation. The assessment is achieved by calculating the Pearson's correlation coefficient and Kendall's $\tau$ between the predicted and observed drug activity. We selected the best models based on the cross validation process (Fig. 2e, Supplementary Fig. a, b). For the 49 protein-coding gene-based models of FDA approved agents, we used the same training procedures as lncRNA-based models.

**Model independent-validation**. To assess the robustness of our pan-cancer as well as cancer-specific models, we obtained drug response data in the CCLE and CTRP. After mapping the cell lines and agents to those in our study, we got 389 overlapping cell lines and 14 overlapping agents for CCLE and 353 cell lines and 76 compounds for CTRP. The analysis is performed on IC50 in CCLE and AUC in CTRP. The prediction performance is evaluated by the Pearson's correlation between predicted and real IC50s in CCLE study and predicted and real AUCs in CTRP (Fig. 3d, e, Supplementary Fig. 3b).

**Prediction of drug response in patients**. Expression of 2614 cancer-related lncRNAs in 3814 TCGA patients with survival information available and was obtained from MiTranscriptome[11]. Patients with stage-1 disease are further filtered out except for the LAML patients. Using the expression data, we constructed an expression matrix $E \in R^{N,p}$, where $N$ is the number of patients and $p$ is the number of lncRNAs. For compound $i$, the predicted response $P_i \in R^{N,1}$ is calculated by the model based on lncRNA expression $e_{N,20} \in E^{N,p}$, forming a final matrix of predicted response $P \in R^{N,265}$. The predicted response is then sorted by values, from which patients of first quantile are labeled as 'sensitive response'. The patients are then categorized by $c$ cancer types, where $\sum_{j=1}^{c} C_j = N$. The sensitive percentage $S_{j,i}^{\text{percent}}$ for compound $i$ is calculated by $\frac{n}{C_j}$, where $n$ is the number of patients that have 'sensitive response' in cancer $j$. Finally, a matrix of sensitive percentage $S_{j,265}^{\text{percent}}$ for all the agents is constructed based on these results (Fig. 4a and Supplementary Fig. 4b).

**Survival analysis**. Univariate Cox regression: Survival information of TCGA patients, including overall survival (OS) and progression free interval (PFI), was obtained from TCGA database. Cox regression based on predicted drug response $P \in R^{N,i}$ was then applied for each compound $i$, where $i \in \{1,2,\dots, 265\}$. The regression algorithm is implemented by Lifelines 0.8.0.1 package. The hazard ratios are calculated by exponentiation of the coefficients from the regression models (Fig. 4b).

Multivariate Cox regression. Clinical information about TCGA patients, including age and disease stages at diagnosis, was obtained from TCGA database. For each patient, the age is dichotomized as 'young' and 'old' with a cutoff at 65 years' old. For patients from cancer $c$, the predicted response of $n$ drugs that are approved for this cancer would be assigned ranks based on values. The weighted average $R$ of the ranks for each patient is given as follow:

$$R' = \frac{\sum_{i=1}^{n} w_i R_i}{\sum_{i=1}^{n} w_i},$$

where

$$w_i = \begin{cases} 1.0, i \in \{1\text{st lineagents}\} \\ 0.5, i \in \{2\text{nd lineagents}\} \end{cases}.$$

Next, Kaplan–Meier analysis is performed based on the weighted average ranks and overall survival (OS) and progression free interval (PFI). After that, the weighted average ranks are sorted by ascending and dichotomized as 'sensitive response' (top 30%), 'partial response' (30 ~ 50%), 'partial resistance' (50 ~ 70%), and 'resistance' (bottom 30%). With the survival information and the input factors (age, disease stage and weighted average rank of the predicted response), a multivariate Cox regression is then performed for each cancer type. The hazard ratios for each of the factors are calculated by exponentiation of the coefficients from the regression models (Fig. 4c).

**Survival analysis on patient with real treatment record**. Patients with real treatment record are segregated by the median of predicted drug response. The analysis is performed on overall survival and a hazard ratio will be given by the univariate cox regression using the segregated drug response. The same procedure is also applied on the predicted drug response by protein-coding gene-based models.

**Identification of multi-drug-response related lncRNAs**. To identify MDR-related lncRNAs that are independent from drug mechanism, we constructed a vector $D$ with length $m$ for each predictive lncRNA $i$. Each element $D_j$ in $D$ denotes the target pathway of the corresponding agent $j$ that lncRNA $i$ is predictive to, and $j \in \{1, 2, \dots, m\}$. In total, $D$ will be expected to have $n$ unique elements, denoted by $C$. Next, for each lncRNA $i$, we calculate the Shannon entropy[63] $H_i$ of $D$ using the following formula:

$$H_i(D) = -\sum_{k=0}^{n} p_{C_k} \log_2 p_{C_k},$$

where

$$p_{C_k} = Pr\left(D_j = C_{k \ j \in \{1,2,\dots,m\}}\right).$$

The resulted entropy metrics will be further transformed into $z$-scores. LncRNAs with a $z$-score >1, i.e., one standard deviation from the right side of the mean, would be selected as an MDR-related lncRNA (Fig. 5a and Supplementary Fig. 5b).

**Co-expression and gene sets enrichment analysis**. We calculated the Pearson's correlation coefficients between 19,680 protein coding genes' expression and 2614 lncRNAs' expression, forming a coefficient matrix $\beta^{p,l}$, where $p$ is the number of protein coding genes and $l$ is the number of lncRNAs. We ranked the protein coding genes based on their expression correlation with lncRNAs. Gene Sets Enrichment Analysis (GSEA) is performed based on cancer hallmarks (h) gene sets and KEGG gene sets from GSEA database[38,64]. The final enrichment score matrix is given by normalized enrichment score (NES) and false discovery rate (FDR) from GSEA. An enrichment with FDR < = 0.25 would be considered as significant enrichment (Supplementary Fig. 5b).

For each target pathway, we construct an lncRNA selection matrix by using top predictive lncRNAs from respective agents. Top predictive lncRNAs are defined as top 20 lncRNAs with highest predictive scores in single agent. An lncRNA selection vector is constructed for each compound, and is merged into a selection-pathway matrix with 21 rows (pathways) and 1292 columns (predictive lncRNAs that are top predictors for at least one compound). Next, one-side Fisher's exact test is performed to assess the enrichment of top lncRNAs in each pathway based on dichotomized enrichment matrix and lncRNA selection matrix (Fig. 6a, Supplementary Fig. 6a).

**Cell culture, RNA interference, and real-time PCR**. Human breast cancer cell lines, Hs578T, BT-474, and MCF-7, and human lung cancer cell line A549 were purchased from American Type Culture Collection (ATCC) and cultured as suggested by ATCC's guidelines. Human ovarian cancer cell line, A2780 and the cisplatin resistant version of the cell line, A2780cis, were obtained from the European Collection of Cell Cultures (ECACC), supplied by Sigma-Aldrich, and cultured in RPMI 1640 medium supplemented with 2 mM glutamine, 10% FBS, 1% penicillin, and 1% streptomycin; A2780cis cells were also supplemented with 1 µM cisplatin. Phoenix cells were kindly provided by Dr. Wen Xie (University of Pittsburgh) and maintained in Dulbecco's Modified Eagle's Medium supplemented with 10% FBS, 1% penicillin, and 1% streptomycin.

For RNA interference, cells were transfected with 40 nM siRNA targeting *EPIC1*, or control siRNA using Lipofectamine RNAiMAX (Thermo Fisher, #13778150) per the manufacturer's instructions. For quantitative real-time PCR (qRT-PCR) analysis, total RNA was isolated 72 h later using an RNeasy Mini kit (Qiagen, #74104) according to the manufacturer's instructions; cDNAs were synthesized from 0.5 µg of total RNA using a High-Capacity cDNA Reverse Transcription Kit (Applied Biosystems, #4368813). qRT-PCR was performed with Power SYBR Green PCR Master Mix (Applied Biosystems, #4367659) on a QuantStudio 6 Flex Real-Time PCR System (Applied Biosystems). Relative gene expression was determined by $\Delta\Delta$Ct normalized to GAPDH.

The following siRNAs (sense, antisense) were used as previously described:[24] *EPIC1* siRNA_A#, CCUUCAGACUGUCUUUGAAdTdT, UUCAAAGACAGUCUGAAGGdTdT; *EPIC1* siRNA_B#, GCUUUCUCUCGGAAACGUGdTdT, CACGUUUCCGAGAGAAAGCdTdT; *EPIC1* siRNA_C#, AGUGUGGCCUCAGCUGAAAdTdT, UUUCAGCUGAGGCCACACUdTdT; control siRNA, GUGCGUUGUUAGUACUAAUdTdT, AUUAGUACUAACAACGCACdTdT. Sequences of primers for qRT-PCR were: *EPIC1* forward, TATCCCTCAGAGCTCCTGCT, and *EPIC1* reverse, AGGCTGGCAAGTGTGAATCT; GAPDH forward, GGTGAAGGTCGGAGTCAACG, and GAPDH reverse, TGGGTGGAATCATATTGGAACA.

**Validation of lncRNA-drug predictive pairs in cell lines.** MCF-7 cells (MCF-7/Vector and MCF-7/EPIC1) and A549 cells (A549/Vector and A549/EPIC1) stably expressing an empty vector and EPIC1 were established with retroviral particles using the previously published method[24]. Briefly, full-length of EPIC1 was inserted into retroviral pBABE-lnc vector with AgeI and XhoI enzymes and the resulting construct was named as lnc-EPIC1. To establish stable EPIC1-expressing cells, 10 μg of pBABE-lnc or lnc-EPIC1 plasmids were transfected into a 10-cm culture dish of Phoenix cells to produce retroviral particles, and retroviruses were collected 48 h post transfection. Then, cells were transduced for 24 h with the retroviruses and selected with puromycin.

The ectopic expression level of EPIC1 in stable cells was confirmed using PCR. To validate lncRNA-Drug interactions, EPIC1 knockdown and overexpressed cells were seeded at 2000 cells per well in 96-well culture plates and incubate for overnight at 37 °C, 5% CO₂. After treatment with a series of twofold diluted drugs (JQ-1 and I-BET-762) for 48 h, MTT assays were performed with a CellTiter 96 Non-Radioactive Cell Proliferation Assay Kit (Promega, #G410) following the manufacture's guidelines. The absorbance value was measured at 570 nm using an xMark Microplate Spectrophotometer (Bio-Rad) with a reference wavelength of 630 nm and the IC50 of JQ-1 and I-BET-762 on cells was calculated, respectively.

**Code availability.** Source code for training and testing the LENP model is available at https://figshare.com/articles/Elastic_net_regression_training_and_testing_LENP_/6480461. The authors declare that all the other scripts generating the figures and supporting the findings of this study are available from the corresponding author upon reasonable request.

**Data availability.** RNA-seq of EPIC1 knockdown cell lines can be obtained from Gene Expression Omnibus (GEO; http://www.ncbi.nlm.nih.gov/gds) under accession number GSE98538 in a pre-processed format. The drug response data of 265 agents in 1001 cancer cell lines were downloaded from GDSC database (http://www.cancerrxgene.org). The gene expression matrix of cancer cell lines was downloaded from Expression Atlas (http://www.ebi.ac.uk/gxa/home). The gene expression matrix of TCGA patients was downloaded from MiTranscriptome (http://mitranscriptome.org). The authors declare that all the other data supporting the findings of this study are available in the article and its Supplementary Data files or from the corresponding authors upon reasonable request.

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

## Acknowledgements

This study was supported by a grant from the Shear Family Foundation (to D.Y.), a Career Development Award of RPCI-UPCI Ovarian Cancer SPORE (P50 CA159981; to D.Y.), a grant from the Elsa U. Pardee Foundation (to D.Y.), and R01CA223788 (to S.L.). This project is supported in part by award P30CA047904. We thank the Center for Simulation and Modeling (SaM) at the University of Pittsburgh for computing assistance. We also thank Drs. Anil Sood and Han Liang for critical reading.

## Author contributions

Conceptualization, D.Y., M.Z.; Methodology, Y.W., M.Z., and D.Y.; Formal analysis, Y. W., M.Z.; Investigation, Y.W., M.Z., Z.W., J.L., J.X., S.L., and D.Y.; Resources, M.Z. and D.Y.; Writing—original draft, Y.W., M.Z., and D.Y.; Writing—review & Editing, Y.W., M.Z., and D.Y.; Supervision, M.Z., and D.Y.; Funding acquisition, D.Y.

## Additional information

**Competing interests:** The authors declare no competing interests.

