## [Peer Review File · Nature Communications]

Reviewers' comments:

Reviewer #1 (Remarks to the Author):

The manuscript entitled "Systematic Identification of Non-coding Pharmacogenomic Interactions in Cancer" is very interesting, original and novel study which uses mostly publically available data for a more in-depth and sophisticated analysis. The results are overall of interest, although some of the correlations are not very strong which could be discussed in more detail. Importantly, a prediction for one lncRNA - drug interaction was experimentally verified. Nonetheless, several important issues need to be addressed prior to publication.

Major issues:

1) Specificity

The number of more than 160.000 lncRNA-drug interactions, i.e. more than 600 lncRNAs for each of the 265 drugs on average, gives rise to questions about the specificity of this analysis. This could be remedied by an in-depth analysis of the robustness of these predictions under different stringencies of the analysis and best if additional predictions would be experimentally verified, e.g. linc09992 and others. Also, more explanation of the CCLE analysis (figure 3d,e) would be desirable how many of the individual lncRNA-drug interactions for the 14 overlapping drugs were actually found to be significant in both groups and how many were found only in one of the groups?

2) Patient drug response

The analysis of the tumor samples and drug response are mostly based on assumptions on how the patients could have been treated. Since data is available for several cohorts including the drug treatment information, the authors should look for such a cohort to verify their findings with real data.

3) EPIC1

The authors state that EPIC1 was "upregulated" in MCF-7 cells (l. 314) - nonetheless, the authors use this cell line for overexpression experiments. These experiments need to be repeated in non-EPIC1-expressing cells - and vice versa, proper knockdown experiments (with at least three independent siRNAs) should be performed in EPIC1-expressing cells to test for drug sensitivity (not only gene expression) in multiple cell lines of each group. Next, the ectopic overexpression and knockdown levels of EPIC1 should be compared to the endogenous differences between sensitive and resistant cell lines.

Minor issues:

4) Statistics

In line 240/241, p-values of $p=0.05$ and $p=0.08$ are called significant, in lines 274/275 $p=0.065$ or $p=0.178$ or lines 310/311 $p=0.067$ and $p=0.052$. Why?

5) Figures

The figures are generally very dense or overloaded - more space between the panels would be desirable.

Reviewer #2 (Remarks to the Author):

NCOMMS-18-00096

Wang et al. Systematic identification of non-coding pharmacogenomic interactions in cancer

The authors utilized a set of bioinformatics tools to build up a genome wide correlation map between long non-coding RNAs (lncRNAs) and drug response in cancer. The experimental data they used are published data containing information of 11,950 lncRNAs, 265 anticancer agents, 27 cancer types obtained in 5,605 tumors and 1,005 cancer cell lines. The analysis is very comprehensive. The generated results are convincing readers to believe the identified correlation is solid, which has not been discovered before. The authors further used experimental approaches to validate a top predictive lncRNA and provided evidence to show their findings are informative and useful for identification of lncRNAs as biomarkers for prediction of drug response in cancer treatment. The manuscript is timely important for the wider field related to non-coding RNAs and cancer. A few minor concerns may help the authors to further improve the quality of the manuscript.

Minor concerns:

1. In the abstract, introduction, and discussion section, the author described two phenotype outcomes: "drug response" and "drug resistance". Although they have similar meaning in some degree, this might cause confusion to readers. Phenotypes of "drug response" and "drug resistance" should be clearly defined with measurable characters.

2. Non-coding RNAs include both long non-coding RNAs and miRNAs. Other researches also showed that miRNAs can be used as biomarkers to predict drug response to cancer. What are the advantage and disadvantage using lncRNAs as biomarkers in comparison to miRNAs?

3. In the introduction part (line 77-78), the authors mentioned one of the specific aims is to find "the underlying mechanisms of lncRNA-mediated cell line response to anti-cancer agents". The authors used HOTAIR as an example in the later experiment. So far, a few mechanisms of lncRNAs have been discovered, such as lncRNA's functions as signals, decoys, guides, and scaffolds. However, in the later experiments the authors did not include any specific experiments to discover the mechanisms of lncRNAs in gene regulation. This specific aim might not be completed in the current manuscript.

4. In the first experiment, the author picked three parameters of lncRNAs to correlate with cancer: expression, copy number, and DNA methylation. The first two are easy to understand, however the DNA methylation status is a little confusing. Did the authors measure the DNA methylation status of the promoter regions of selected lncRNAs? DNA

methylation is usually related to repressed gene expression. In Figure 1d, the result seemed to show that low methylation status correlates to low expression shown in Figure 1c.

5. In the experiment testing drug response, there are several well-accepted mechanisms in causing anticancer drug resistance, including drug efflux, cell death inhibition, and drug inactivation. In later study, the authors correlated lncRNAs with drug metabolizing enzymes. The correlation data between targeted lncRNAs and some important transporters and apoptosis related genes might be added to make the results more convincing, such as the correlation of expression between lncRNAs and efflux transporters.

6. The authors mentioned "agents targeting the same pathway tended to share similar predictive lncRNAs (row 153-154)". Does this phenomenon relate to same drug transporters or drug metabolizing enzymes?

7. The authors identified an lncRNA, LINC00992, as a potential regulator of CYP genes. The authors referred two references (29 and 33) to indicate they play important roles in chemotherapy resistance in cancer. However, reference 29 is a review article and reference 33 has no referred information.

8. In line 81-82, the sentence "To our best knowledge, this is the first study ..." is duplicated.

9. Several places have grammar errors. In line 116, "data includes" should be "data include". In line 292, "iBETs achieves" should be "iBETs achieve". In line 365, "This serve as ..." should be "This serves as ...". In line 528, "Expression level of these lncRNAs are ..." should be "Expression levels of these lncRNAs are ...". In line 688-691, "Next, one-sided (greater) Fisher's exact test is ..." needs to be rewritten.

Reviewer #3 (Remarks to the Author):

In this study, Wang et al. performed a computational systematic prediction and analysis of non-coding pharmacogenomic interactions in cancer. This study has potential value in addressing cancer drug sensitivity and resistance but serious problems exist in current version.

1. The authors used KNN algorithm for matching the tissue of origin of cell lines to primary tumors. There are some serious problems. First, the authors did not correctly understand the algorithm of KNN and the application of KNN in this paper is totally wrong. KNN works for assigning the given sample to the most frequent sample class for the top K nearest neighbors. However, here the authors just considered if the k-th nearest neighbors is the correct or not? I don't think this procedure is correct and feasible. Here is a simple example, in the first line of the suppl table 1. For the top five nearest neighbors, KNN should predict the input as LUAD because there are two LUAD samples but the authors predict the input as PAAD because the 5-th sample is PAAR. This procedure is wrong. Second, the percentage of successful identification is not high. I am wondering how about the result if the authors

combining the three features, expression, methylation, and copy number.

2. The so-called "lncRNA-drug interactions" are not real physical interactions between lncRNAs and drugs but are some functional interaction. So the term "lncRNA-drug interactions" is not such correct.

3. I cannot find the AUC data in Supplementary Table 2a, as the authors stated "For each cell line, the drug response data includes the values of IC50 and area under the curve (AUC) of 265 anti-cancer agents from the GDSC database".

4. In Supplementary Table 2b, I found 114,575 "lncRNA-drug interactions" but not as the authors stated "103,155" in the main document. Moreover, I cannot understand why the number of unique lncRNA-drug interactions in total (162,327) is greater than that in cancer-specific models (103,155). According to my understanding the unique one should be less.

5. Is the lncRNA-based LENP model better than previous other molecular based models? I did not find a comparison.

Summary of the revision: We thank the reviewers for their comments on the strengths of the manuscript. We are also indebted to the reviewers for their extremely insightful suggestions to help us further improve this study. In the revised manuscript, we have updated 13 new panels to the main figures and 14 new panels to the supplemental figures. The major changes of the manuscript are summarized below:

- 1) We have included two independent cancer cell line pharmacogenomics databases (i.e., CTRP and CCLE) to determine the specificity of identified lncRNA-drug predictive pairs and validate the performance of our prediction model.
- 2) We have parsed and curated 10,237 TCGA patients' chemotherapy history from raw clinical records. With the drug treatment history of patients, we are now able to apply our LENP models to only patients that were treated with the corresponding agents and evaluate our model performance.
- 3) We have done a comprehensive analysis between lncRNA and drug transporter and apoptosis pathway to explore lncRNA's regulation of drug metabolism.
- 4) We have built protein coding genes-based drug response prediction models using same algorithm and compared their performances with lncRNA-based models in cancer cell lines and patients.
- 5) Finally, we have used three *EPIC1* siRNAs to knockdown *EPIC1* in three cancer cells (i.e., MCF-7, BT-474, ZR-75-1) and overexpressed *EPIC1* in low endogenous *EPIC1* cells (i.e., A549) to determine the *EPIC1* regulation of iBETs response in multiple cell lines.

In addition to the aforementioned major revisions, we have performed additional revisions to comprehensively address all the concerns/comments from the reviewers. We have highlighted the specific changes in the manuscript.

Point-to-point responses to all of the reviewers' original comments:

Reviewer #1 (Remarks to the Author):

The manuscript entitled "Systematic Identification of Non-coding Pharmacogenomic Interactions in Cancer" is very interesting, original and novel study which uses mostly publically available data for a more in-depth and sophisticated analysis. The results are overall of interest, although some of the correlations are not very strong which could be discussed in more detail.

Importantly, a prediction for one lncRNA - drug interaction was experimentally verified.

Nonetheless, several important issues need to be addressed prior to publication.

Major issues:

1) Specificity

The number of more than 160,000 lncRNA-drug interactions, i.e. more than 600 lncRNAs for each of the 265 drugs on average, gives rise to questions about the specificity of this analysis. This could be remedied by an in-depth analysis of the robustness of these predictions under different stringencies of the analysis and best if additional predictions would be experimentally verified, e.g. linc09992 and others.

Also, more explanation of the CCLE analysis (figure 3d,e) would be desirable how many of the individual lncRNA-drug interactions for the 14 overlapping drugs were actually found to be significant in both groups and how many were found only in one of the groups?

Response: To address the specificity of the identified lncRNA-drug predictive pairs, we have used two independent datasets (i.e., CCLE and Cancer Therapeutics Response Portal [CTRP]) to determine the robustness of our analysis strategy under different stringencies. We calculated the correlation between the lncRNA expression and the drug response between predictive lncRNA-drug pairs and non-predictive lncRNA-drug pairs in CCLE and CTRP databases. The specificity is defined as the distribution distance between the correlation of lncRNA expression

and drug response in predictive or non-predictive lncRNA-drug pairs in these two independent datasets. As shown in **Supplementary Fig. 2a**, the specificity of the predictive lncRNA-drug pairs increases as the predictive score (PS) cutoff becomes more stringent. The specificity of the predictive lncRNA-drug pairs reaches to a turning point at PS cutoff = 0.25, where we can achieve a good balance of specificity and sensitivity. Therefore, we have now identified 27,341 pan-cancer predictive lncRNA-drug pairs showing $PS \geq 0.25$, with a median of 100 predictive lncRNAs per agent. If we used Spearman correlation to select the lncRNA-drug predictive pairs, each drug would have a median of 396 predictive lncRNA (with an FDR < 0.25 as cutoff). Comparing to Spearman correlation analysis, the elastic net algorithm combined with bootstrapping has largely removed the redundancy and increased the specificity by considering the collinearity between features. We have revised methods based on the reviewer's suggestions at **Online Method line 80-90** and revised results in **Fig. 2c, d and e**, **Supplementary Fig. 2a, b and d**, **Fig. 5a, b, c and e**, and **Supplementary Fig. 5b**.

To further determine the specificity of lncRNA-drug predictive pairs, we performed same feature selection procedure in the CCLE and CTRP databases. This analysis is only performed to the shared drugs and cell lines with GDSC, i.e. 14 compounds and 289 cell lines from CCLE and 76 compounds and 353 cell lines from CTRP (**Supplementary Table 2e**). For the 14 overlapping drugs between CCLE and GDSC, we identified 512 and 1,366 lncRNA-drug pairs respectively. Among the 512 lncRNA-drug pairs in CCLE, 90 (17.6%) of them were also identified in GDSC (odds ratio = 5.71, $p = 1.41 \times 10^{-34}$, Fisher's exact test). For the 76 overlapping drugs between CTRP and GDSC, we identified 4,827 and 7,938 lncRNA-drug pairs respectively. Among the 4,827 lncRNA-drug pairs in CTRP, 612 (12.7%) were also found to be significant ($PS \geq 0.25$) in GDSC (odds ratio = 3.59, $p = 8.89 \times 10^{-136}$, Fisher's exact test). Again, the lncRNA-drug pairs identified by LENP have a significantly higher robustness among independent databases than those identified by spearman correlation (3.6% for CCLE and 1.4%

for CTRP). These results have been included in the revised manuscript at **line 135-150** and **Supplementary Fig. 2a**.

2) Patient drug response

The analysis of the tumor samples and drug response are mostly based on assumptions on how the patients could have been treated. Since data is available for several cohorts including the drug treatment information, the authors should look for such a cohort to verify their findings with real data.

Response: Following the reviewer's suggestion, we have parsed and curated the 10,237 TCGA patients' chemotherapy history from raw TCGA clinical records (**Online Method, line 21-24**). Although most cancer patients' chemotherapy treatment data are missing, several cancer types, including breast, ovarian, uterine, and gastric cancer, have relatively complete chemotherapy treatment history in record. We found LENP can predict the therapeutic outcomes for a number of drugs. For example, there are 24 ovarian cancer patients and 210 breast cancer patients receiving tamoxifen treatment in TCGA cohorts. Among those patients, poor survival is observed for patients with higher predicted IC50 by LENP-tamoxifen model (log-rank test: $p = 0.01$ for OV, $p = 0.19$ for BRCA) (**Fig. 4c, Supplementary Fig. 4c, Online Methods**). A trend of poorer survival were observed among patients with higher predicted paclitaxel IC50 in 111 breast cancer patients, 138 ovarian cancer patients and 47 endometrial cancer patients who have been treated with paclitaxel (log-rank test: $p = 0.12$ for BRCA, $p = 0.30$ for OV, $p = 0.10$ for UCEC) (**Fig. 4d, Supplementary Fig. 4e, Online Methods**). In addition, we applied LENP-5FU model to 49 fluorouracil-treated stomach adenocarcinoma patients. We found that patients with lower predicted IC50 tend to have a better survival outcome compared to the rest (log-rank test: $p = 0.08$, STAD) (**Fig. 4d, Online Methods**). These results have been included in **line 250-266, Fig. 4d** and **Supplementary Fig. 4e**.

3) *EPIC1*

The authors state that *EPIC1* was "upregulated" in MCF-7 cells (l. 314) - nonetheless, the authors use this cell line for overexpression experiments. These experiments need to be repeated in non-*EPIC1*-expressing cells - and vice versa, proper knockdown experiments (with at least three independent siRNAs) should be performed in *EPIC1*-expressing cells to test for drug sensitivity (not only gene expression) in multiple cell lines of each group. Next, the ectopic overexpression and knockdown levels of *EPIC1* should be compared to the endogenous differences between sensitive and resistant cell lines.

Response: Following the reviewer's suggestion, we have included another cell line A549, which does not express endogenous *EPIC1* (**Fig. 6d** and **Supplementary Fig. 7a**). As shown in **Supplementary Fig. 7e**, overexpression of *EPIC1* in A549 can significantly increase JQ1 resistance. Three independent *EPIC1* siRNAs were used to knockdown *EPIC1* in three *EPIC1*-expressing cells (MCF-7, BT-474 and ZR-75-1 cells). Our data indicated that *EPIC1* knockdown could increase the sensitivity of tumor cells to iBETs (**Fig. 6e**, **Supplementary Fig. 7d**). We have also shown that the cell lines, which have similar endogenous *EPIC1* expression level to those in *EPIC1*-knocked-down cells, have comparable IC50s to *EPIC1* knockdown cells. These data have been included in the revised manuscript at **Fig. 6e-f**, **Supplementary Fig. 7d-f** and **line 353-361**.

Minor issues:

4) Statistics

In line 240/241, p-values of $p=0.05$ and $p=0.08$ are called significant, in lines 274/275 $p=0.065$ or $p=0.178$ or lines 310/311 $p=0.067$ and $p=0.052$. Why?

Response: Thanks for pointing out this problem. We have corrected these mistakes in the revised manuscript at lines 246-249 and 348-350.

5) Figures

The figures are generally very dense or overloaded - more space between the panels would be desirable.

Response: Thanks for the suggestion. We have re-arranged our figures in the revised manuscript.

Reviewer #2 (Remarks to the Author):

NCOMMS-18-00096

Wang et al. Systematic identification of non-coding pharmacogenomic interactions in cancer

The authors utilized a set of bioinformatics tools to build up a genome wide correlation map between long non-coding RNAs (lncRNAs) and drug response in cancer. The experimental data they used are published data containing information of 11,950 lncRNAs, 265 anticancer agents, 27 cancer types obtained in 5,605 tumors and 1,005 cancer cell lines. The analysis is very comprehensive. The generated results are convincing readers to believe the identified correlation is solid, which has not been discovered before. The authors further used experimental approaches to validate a top predictive lncRNA and provided evidence to show their findings are informative and useful for identification of lncRNAs as biomarkers for prediction of drug response in cancer treatment. The manuscript is timely important for the wider field related to non-coding RNAs and cancer. A few minor concerns may help the authors to further

improve the quality of the manuscript.

Minor concerns:

1. In the abstract, introduction, and discussion section, the author described two phenotype outcomes: “drug response” and “drug resistance”. Although they have similar meaning in some degree, this might cause confusion to readers. Phenotypes of “drug response” and “drug resistance” should be clearly defined with measurable characters.

Response: Thanks for pointing this out. We have cleaned up the terminology in the revised manuscript. Specifically, the term ‘drug response’ includes both drug resistance and drug sensitivity. When an lncRNA's high (low) expression associates with high IC50, we would define this lncRNA to predict the drug resistance (sensitivity).

2. Non-coding RNAs include both long non-coding RNAs and miRNAs. Other researches also showed that miRNAs can be used as biomarkers to predict drug response to cancer. What are the advantage and disadvantage using lncRNAs as biomarkers in comparison to miRNAs?

Response: We have included a discussion about the disadvantage and advantage of using lncRNAs as biomarkers in comparison to miRNAs in the revised manuscript at **lines 414-418**.

3. In the introduction part (line 77-78), the authors mentioned one of the specific aims is to find “the underlying mechanisms of lncRNA-mediated cell line response to anti-cancer agents”. The authors used HOTAIR as an example in the later experiment. So far, a few mechanisms of lncRNAs have been discovered, such as lncRNA’s functions as signals, decoys, guides, and scaffolds. However, in the later experiments the authors did not include any specific experiments to discover the mechanisms of lncRNAs in gene regulation. This specific aim might not be completed in the current manuscript.

Response: We agree with the reviewer that the mechanism by which the lncRNA regulate gene expression could not be completed in the current manuscript. We have thus removed that aim from the introduction part, **line 79-80**. Regarding the mechanism of the lncRNA we experimentally validated in the later experiment, we are happy to update that our recently publication indicate that this lncRNA can directly interact with MYC protein and enhance its transcriptional activity (reference [24]). The anti-cancer effect of BET inhibitor has been documented to partially rely on its inhibition of MYC expression. We speculate that this lncRNA may lead to BET inhibitor resistance through enhancing the MYC transcriptional activity and discuss about this at **line 438-443**. As the reviewer suggest, we will comprehensively investigate this hypothesis in future studies.

4. In the first experiment, the author picked three parameters of lncRNAs to correlate with cancer: expression, copy number, and DNA methylation. The first two are easy to understand, however the DNA methylation status is a little confusing. Did the authors measure the DNA methylation status of the promoter regions of selected lncRNAs? DNA methylation is usually related to repressed gene expression. In Figure 1d, the result seemed to show that low methylation status correlates to low expression shown in Figure 1c.

Response: Yes, we measured the lncRNA methylation status in lncRNA promoter regions using our published method (reference [24]). In **Fig. 1d**, we only showed the pairwise correlation of the methylation profile between cancer cell lines and cancer patients. In **Fig. 1c**, we only compared the expression profile between cancer cell lines and patients. We did not include the comparison between methylation and expression in these analyses. To address the reviewer's concern about the correlation between DNA methylation and gene expression, we have performed the correlation analysis between the lncRNA expression and their promoter methylation status. This analysis indicates that lncRNA expression is indeed negatively associated with their promoter methylation status (reference [24]) (**Supplementary Fig. 1c**).

5. In the experiment testing drug response, there are several well-accepted mechanisms in causing anticancer drug resistance, including drug efflux, cell death inhibition, and drug inactivation. In later study, the authors correlated lncRNAs with drug metabolizing enzymes. The correlation data between targeted lncRNAs and some important transporters and apoptosis related genes might be added to make the results more convincing, such as the correlation of expression between lncRNAs and efflux transporters.

Response: We are also very interested in the association between multi-drug-response related lncRNAs and drug transporters. We have extended this analysis to more KEGG pathways. The ABC transporters ($p = 2.01 \times 10^{-16}$) and apoptosis related genes ($p = 4.11 \times 10^{-57}$) also showed a good correlation with multi-drug-response related lncRNAs. These results have been included in **line 294-298**. Thank the reviewer for this suggestion.

6. The authors mentioned “agents targeting the same pathway tended to share similar predictive lncRNAs (row 153-154)”. Does this phenomenon relate to same drug transporters or drug metabolizing enzymes?

Response: Following reviewer’s suggestion, we have performed a similar analysis using the drug transporters and drug metabolizing enzymes that have been established to regulate a specific drug’s PK in cancer cell according to PharmGKB database (www.pharmgkb.org). We compared the predictive lncRNAs among 17 agents that are validated substrates of ABCB2, but found no significance in sharing similar predictive lncRNAs comparing to the random expectation ($p = 0.23$, two-sample T test).

7. The authors identified an lncRNA, LINC00992, as a potential regulator of CYP genes. The authors referred two references (29 and 33) to indicate they play important roles in chemotherapy resistance in cancer. However, reference 29 is a review article and reference 33 has no referred information.

Response: We are sorry for this oversight. We have corrected the citation (**line 318**) in this revision.

8. In line 81-82, the sentence “To our best knowledge, this is the first study ...” is duplicated.

Response: Thanks for pointing it out. We have removed the duplicates in this revision.

9. Several places have grammar errors. In line 116, “data includes” should be “data include”. In line 292, “iBETs achieves” should be “iBETs achieve”. In line 365, “This serve as ...” should be “This serves as ...”. In line 528, “Expression level of these lncRNAs are ...” should be “Expression levels of these lncRNAs are ...”. In line 688-691, “Next, one-sided (greater) Fisher’s exact test is ...” needs to be rewritten.

Response: Thanks for pointing it out. We have corrected these sentences and cleaned up the grammar mistakes in this revision at **line 114, 333, 437** in manuscript and **line 192-195** in **Online Methods**.

Reviewer #3 (Remarks to the Author):

In this study, Wang et al. performed a computational systematic prediction and analysis of non-coding pharmacogenomic interactions in cancer. This study has potential value in addressing cancer drug sensitivity and resistance but serious problems exist in current version.

1. The authors used KNN algorithm for matching the tissue of origin of cell lines to primary tumors. There are some serious problems. First, the authors did not correctly understand the algorithm of KNN and the application of KNN in this paper is totally wrong. KNN works for assigning the given sample to the most frequent sample class for the top K nearest neighbors. However, here the authors just considered if the k-th nearest neighbors is the correct or not? I don't think this procedure is correct and feasible. Here is a simple example, in the first line of the supplementary table 1. For the top five nearest neighbors, KNN should predict the input as LUAD

because there are two LUAD samples but the authors predict the input as PAAD because the 5-th sample is PAAR. This procedure is wrong. Second, the percentage of successful identification is not high. I am wondering how about the result if the authors combining the three features, expression, methylation, and copy number.

Response: Thank the reviewer for helping us identify this oversight. In the revised manuscript, we assess the consistency of genomic alterations between primary tumors and cancer cell lines using a previously published method (reference [22]), which is more like a “nearest neighbor matching” method. This method simply determines if there is correct cancer type matching within the top k nearest neighbors (see **Online Method, line 42-45**). This analysis indicates that cell lines could realistically recapitulate the lncRNA alterations in patient tumor. Within the top 5 nearest neighbors, the algorithm could 100% correctly match the tissue of origin of cell lines to primary tumors using lncRNA expression with a random expectation of matching rate at 33.3%. This percentage is around 89.5% when using methylation and 88.9% when using copy number with random expectation at 15.8% and 27.8%, respectively.

Following the suggestion of reviewer, we have further combined the three features, i.e. expression, methylation, and copy number (see **Online Method, line 35-46**), and the percentage of successful identification within the top 5 neighbors is around 94.4%, which is lower than that of expression (100%) but higher than those of copy number (80.6%) and methylation (89.5%). The results have been included in **Fig. 1f, line 100-109**.

2. The so-called "lncRNA-drug interactions" are not real physical interactions between lncRNAs and drugs but are some functional interaction. So the term "lncRNA-drug interactions" is not such correct.

Response: We agree with the reviewer that “interaction” is not appropriate in the context of this manuscript. We have thus changed the “lncRNA-drug interactions” to “lncRNA-drug predictive pairs” in the revised manuscript.

3. I cannot find the AUC data in Supplementary Table 2a, as the authors stated "For each cell line, the drug response data includes the values of IC50 and area under the curve (AUC) of 265 anti-cancer agents from the GDSC database".

Response: Sorry for this oversight. We have included it in the revised manuscript in **Supplementary Table 2b**.

4. In Supplementary Table 2b, I found 114,575 "lncRNA-drug interactions" but not as the authors stated "103,155" in the main document. Moreover, I cannot understand why the number of unique lncRNA-drug interactions in total (162,327) is greater than that in cancer-specific models (103,155). According to my understanding the unique one should be less.

Response: The “162,327” unique lncRNA-drug pairs contain predictive pairs from both pan-cancer and cancer-specific models, while the “103,155” pairs only include those from cancer-specific models. We agree with the reviewer that this description is misleading and have carefully revised them in the revised manuscript. In the revised manuscript (**line 130-132**), we have also included two independent datasets (i.e., CCLE and Cancer Therapeutics Response Portal [CTRP]) to optimize the cutoff of identifying lncRNA-drug predictive pairs (**Supplementary Fig. 2a**). Now, we have 27,341 lncRNA-drug predictive pairs in pan-cancer analysis, and 34,336 lncRNA-drug pairs in cancer-specific analysis. These results have been included in the revised manuscript at **line 130-132, Supplementary Fig. 2a**.

5. Is the lncRNA-based LENP model better than previous other molecular based models? I did not find a comparison.

Response: A previously published work (reference [2]) has built Elastic Net regression model using protein-coding genes (PCGs) to predict drug response in cell lines. Here, we constructed

the PCG-based models for 49 FDA approved agents using the same training-testing framework as the one used in that work. We compared the performance of PCG and LENP models in predicting cell line response and the patient survival outcome, and we observed an overall comparable performance between LENPs and PCG in both scenarios. Interestingly, the LENP models can outperform PCG-based models in many patient cases. For example, LENP-tamoxifen model could better predict the prognosis of OV patients treated with tamoxifen (log-rank test: $p = 0.01$; hazard ratio (HR), 3.62; 95% confident interval (95%CI), 1.26-11.13) than PCG-based model (log-rank test: $p = 0.13$; HR, 2.32; 95%CI, 0.74- 7.24). The predicted 5FU resistance by LENP model is better associated with poor survival of 5FU-treated STAD patients (log-rank test: $p = 0.07$; HR, 2.24; 95%CI, 0.89-5.64) than that of PCG-based model (log-rank test: $p = 0.66$; HR, 1.23; 95%CI, 0.49-3.09). In the cases of paclitaxel-treated OV patients and UCEC patients, the LENP models predicted resistant patients would undergo a trend of poor prognosis (OV: HR, 1.28, 95%CI, 0.80-2.07; UCEC: HR, 5.01, 95%CI, 0.58-42.99), however the PCG-based models predicted resistant patients would undergo a trend of beneficial prognosis (OV: HR, 0.65, 95%CI, 0.39-1.08; UCEC: HR, 0.18, 95%CI, 0.02-1.58). These results have been included in the revised manuscript at **lines 267-281, Supplementary Fig. 4f and 4g**.

On top of the training of novel drug prediction models, our feature selection strategy has also identified lncRNA-drug predictive pairs, which may help reveal novel lncRNA regulators for drug response in cancer. We have included a discussion about this in the revised manuscript at **line 401-405**. We thank the reviewer's suggestion to help us improve the study.

REVIEWERS' COMMENTS:

Reviewer #1 (Remarks to the Author):

The authors have appropriately responded and addressed my previous concerns. Only one editorial change would be needed from my perspective:

previous 1) Specificity

The number of drug interactions has been strongly decreased from 160.000 to 27.000. Importantly, the authors have also compared different datasets and find significantly enriched overlaps between the predictions. Nonetheless, it should be emphasized that still only between 13% and 18% of interactions are found overlapping between the datasets emphasizing that the specificity of this analysis is likely still low and that the interactions described here are more hypotheses rather than proven interactions. I still think that the dataset is of interest and should get published, but this point of specificity should be clarified in the discussion to make readers and users of the data more aware of it.

Reviewer #2 (Remarks to the Author):

The authors have submitted a revised manuscript to address all my comments very carefully and appropriately. I don't have any further concern.

Reviewer #3 (Remarks to the Author):

In the revised manuscript, the authors answered my questions well and I don't have further comments.

Summary of the revision: We thank the reviewers for their positive comments on the manuscript. In the revised manuscript, we have added a new paragraph in the discussion based on reviewer #1's suggestion. We have highlighted the specific changes in the manuscript.

Point-to-point responses to all of the reviewers' original comments:

Reviewer #1 (Remarks to the Author):

The authors have appropriately responded and addressed my previous concerns. Only one editorial change would be needed from my perspective:

previous 1) Specificity

The number of drug interactions has been strongly decreased from 160.000 to 27.000.

Importantly, the authors have also compared different datasets and find significantly enriched overlaps between the predictions. Nonetheless, it should be emphasized that still only between 13% and 18% of interactions are found overlapping between the datasets emphasizing that the specificity of this analysis is likely still low and that the interactions described here are more hypotheses rather than proven interactions. I still think that the dataset is of interest and should get published, but this point of specificity should be clarified in the discussion to make readers and users of the data more aware of it.

Response: Thanks for the suggestion. We agree with the reviewer's concern about the specificity of our analysis. We have included a discussion about this issue in the revised manuscript at lines 406-418.

Reviewer #2 (Remarks to the Author):

The authors have submitted a revised manuscript to address all my comments very carefully and appropriately. I don't have any further concern.

Response: We thank the reviewer's insightful suggestions to help us improve our study.

Reviewer #3 (Remarks to the Author):

In the revised manuscript, the authors answered my questions well and I don't have further comments.

Response: We thank the reviewer's insightful suggestions to help us improve our study.